# Covalently reactive microparticles imbibe blood to form fortified clots for rapid hemostasis and prevention of rebleeding

Ting Chen[1], Chaonan Xiao[1], Xianjun Chen [2], Ziyi Yang[3,4], Jingwei Zhao[1,3,4], Bingkun Bao [1], Qingmei Zeng[1], Li Jiang[1], Xinyi Huang[1], Yi Yang [2], Qiuning Lin [1,5] ✉, Wei Gong[3,4] & Linyong Zhu [1,5] ✉

Owing to the inherently gradual nature of coagulation, the body fails in covalently crosslinking to stabilize clots rapidly, even with the aid of topical hemostats, thus inducing hemostatic failure and potential rebleeding. Although recently developed adhesives confer sealing bleeding sites independently of coagulation, interfacial blood hampers their adhesion and practical applications. Here, we report a covalently reactive hemostat based on blood-imbibing and -crosslinking microparticles. Once contacting blood, the microparticles automatically mix with blood via imbibition and covalently crosslink with blood proteins and the tissue matrix before natural coagulation operates, rapidly forming a fortified clot with enhanced mechanical strength and tissue adhesion. In contrast to commercial hemostats, the microparticles achieve rapid hemostasis (within 30 seconds) and less blood loss (approximately 35 mg and 1 g in the rat and coagulopathic pig models, respectively), while effectively preventing blood-pressure-elevation-induced rebleeding in a rabbit model. This work advances the development and clinical translation of hemostats for rapid hemostasis and rebleeding prevention.

Hemorrhage is a major cause of mortality, accounting for more than four million deaths worldwide annually[1]. Once bleeding occurs, the body initiates a series of coagulation processes, consisting of a transition from the initial deposition of platelet plugs[2] to physically crosslinked, reversible fibrin aggregates[3,4]. In the final phase of coagulation, active coagulation factor XIII (FXIIIa) mediates covalent crosslinking between fibrin fibers, as well as between fibrin fibers and the tissue matrix. This stabilizes the clots, enhancing their mechanical strength and adhesion to withstand blood pressure and resist fibrinolysis[5–7]. However, this process takes 6 to 10 minutes in humans[5,8], often rendering it ineffective for rapid hemorrhage management.

Topical hemostats are used as adjunctive measures to improve hemostasis[9]. The majority of the existing United States Food and Drug Administration (FDA)-approved topical hemostats aim to accelerate hemostasis by mimicking, amplifying, and/or leveraging the early physical crosslinking process[10,11]. They increase the topical concentration of blood by absorbing water (e.g., microporous polysaccharide[12], gelatin sponges[13], and oxidized cellulose[13]), supplementing coagulation factors (e.g., fibrin and thrombin[14]), or agglutinating blood cells (e.g., chitosan[15]). However, the ultimate covalent crosslinking still depends on the inherently gradual coagulation process. Therefore, these coagulation-based hemostats cannot form stabilized clots with sufficient mechanical strength and tissue adhesion

[1]School of Biomedical Engineering, Shanghai Jiao Tong University, Shanghai, China. [2]Optogenetics & Synthetic Biology Interdisciplinary Research Center, State Key Laboratory of Bioreactor Engineering, East China University of Science and Technology, Shanghai, China. [3]Department of General Surgery, Xinhua Hospital, affiliated to Shanghai Jiao Tong University School of Medicine, Shanghai, China. [4]Shanghai Key Laboratory of Biliary Tract Disease Research, Shanghai, China. [5]These authors jointly supervised this work: Qiuning Lin, Linyong Zhu. ✉e-mail: qiuninglin@sjtu.edu.cn; linyongzhu@sjtu.edu.cn

immediately[16,17], inducing hemostatic failure in cases of massive hemorrhage[18] and coagulopathic patients[19,20]. Even worse, they fail to prevent rebleeding[21,22], which usually occurs many times before definitive hemostasis is achieved and is associated with high morbidity and even mortality[23,24].

A new generation of coagulation-independent hemostatic agents, represented by adhesive sealants, offers a promising alternative to coagulation-based hemostats for achieving rapid hemostasis[11,20]. Studies have highlighted that pre-crosslinked or in situ-crosslinked sealants confer sealing wounds and avoid blood leakage[25–28]. However, a notable challenge remains: interfacial fluids (e.g., blood and mucus) present on the tissue surface hampers adhesion by competing with the tissue surface for adhesive groups on sealants[29,30]. Recently, the introduction of blood-repelling technologies has helped sealants

remove interfacial blood and achieve rapid hemostatic sealing of bleeding tissues[31–33]. However, these achievements often come at the expense of other desirable attributes, such as the need for a hydrophobic matrix[34], additional tools[25,31], or auxiliary application of steady pressure[34,35], as well as the challenge that introducing reactive components may compromise the hemostat's shelf life[33].

Here, inspired by FXIIIa-mediated covalent crosslinking that stabilizes clots at the final phase of natural coagulation, we describe a covalently reactive hemostat based on blood-imbibing and -crosslinking microparticles (BICMs, Fig. 1a). Upon contact with blood, the BICMs automatically mix with blood via imbibition, and covalently crosslink with blood proteins and the tissue matrix, rapidly forming a fortified clot with enhanced mechanical strength and tissue adhesion before natural coagulation operates. The BICMs enable rapid hemos-

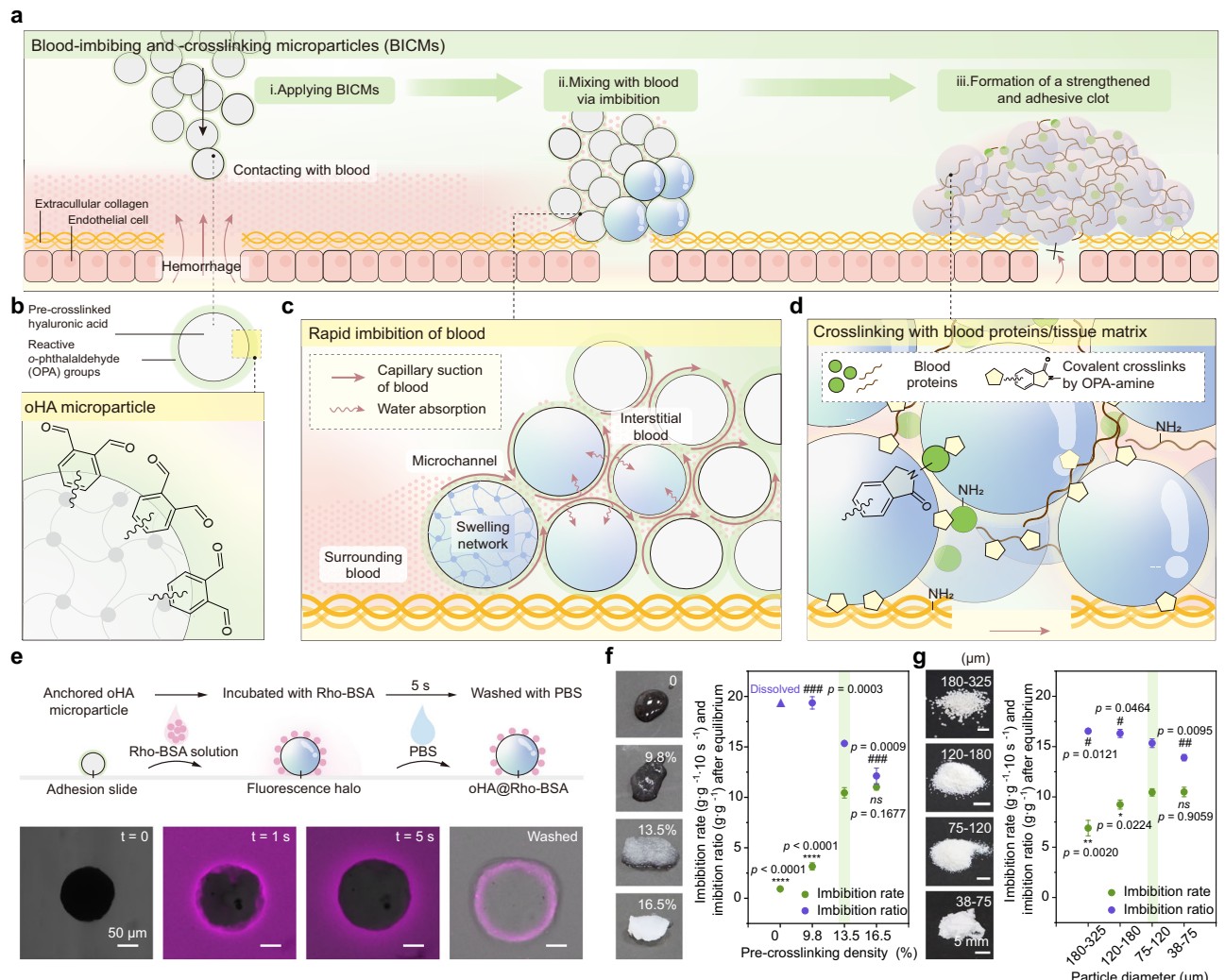

**Fig. 1 | Design and mechanism of the coagulation-independent blood-imbibing and -crosslinking microparticles (BICMs) for rapid hemostasis and prevention of rebleeding. a** Schematic working principle of BICMs at the bleeding site to form a fortified clot with enhanced mechanical strength and tissue adhesion. **b** The structure of a crosslinked hyaluronic acid microparticle modified with *o*-phthalaldehyde (OPA) groups (oHA). **c** Rapid blood imbibition by oHA microparticles via capillary suction and water absorption. **d** The OPA groups on the microparticle surface covalently crosslink with amines present on proteins in imbibed blood and on the tissue matrix. **e** Visualization of the imbibition behavior and reaction between Rhodamine-labeled bovine serum albumin (Rho-BSA, 50 mg mL$^{-1}$, PBS, pH = 7.4) and oHA microparticles. The schematic illustration (top) and microscopy

images (bottom) depict the procedure and results, respectively. **f, g** Effects of (**f**) pre-crosslinking density and (**g**) microparticle size on the imbibition rate (within 10 seconds) and imbibition ratio (after saturation) of oHA microparticles. Data are presented as means ± s.d. (*n* = 3 independent samples for **f** and **g**). All statistical analyses were performed using an unpaired two-tailed Student *t*-test. * *p* < 0.05, ** *p* < 0.01, *** *p* < 0.001, **** *p* < 0.0001 (for comparison of imbibition rate); # *p* < 0.05, ## *p* < 0.01, ### *p* < 0.001, #### *p* < 0.0001 (for comparison of imbibition ratio). Comparison was performed between pre-crosslinking densities of 0, 9.8%, or 16.5% versus 13.5%, as well as between particle size of 180–325 μm, 120–180 μm, or 38–75 μm versus 75–120 μm. Source data are provided as a Source Data file.

tasis and prevent rebleeding, even in coagulopathic conditions. Notably, unlike sealants, the BICMs introduce blood as a reactive, structural component, avoiding competition between blood and the tissue surface for adhesion sites.

## Results

### Design and overview of BICMs

To implement our proposed strategy, we initially fabricated pre-crosslinked hyaluronic acid (HA) microparticles using 1,4-butanediol diglycidyl ether (BDDE) crosslinker (denoted as xHA, Supplementary Fig. 1) owing to their ability to absorb and retain water, as well as their good biocompatibility and biodegradability[36]. To enable the covalent crosslinking with blood proteins independent of coagulation cascade activation (i.e., in the absence of enzymatic reactions or catalysts), we targeted the abundant primary amine groups present on proteins. The reaction between o-phthalaldehyde (OPA) groups and primary amines is rapid and chemoselective for the formation of phthalimidine linkages under physiological conditions[37,38]. In contrast to some widely used reactive groups—such as N-hydroxysuccinimide (NHS) and pentafluorophenyl (PFP) active esters, aromatic and alkyl aldehydes, and catechol groups—OPA groups (0.1 mM) demonstrated higher reactivity. OPA achieved 100% conversion with amines (0.4 mM) after a 60-minute incubation under physiological conditions, which was 2-fold and 7-fold higher than NHS and PFP active esters, respectively (Supplementary Fig. 2a–d). Yet, aromatic and alkyl aldehydes, as well as catechol groups, exhibited no significantly detectable reactivity (1.28%, 0.25%, and 0.99%, respectively). Notably, OPA groups maintained their reactivity after 30 days of storage in phosphate-buffered saline (PBS, pH = 7.4), while the reactivity of active esters declined significantly within three hours (Supplementary Fig. 2e). These results revealed that OPA groups exhibit both high reactivity toward primary amines and exceptional stability in aqueous solution. Afterward, we grafted OPA moieties onto xHA microparticles via an amide coupling reaction to obtain OPA-modified HA (oHA) microparticles (Supplementary Figs. 3–5).

The resulting oHA microparticles feature reactive OPA-modified exteriors and pre-crosslinked, absorbent hyaluronic acid networks (Fig. 1b). In principle, once administered, the oHA microparticles surrounding the bleeding site should imbibe nearby blood into their interstitial microchannels between the microparticles through capillary suction. The oHA microparticles then retain water from the imbibed blood within their pre-crosslinked networks. Consequently, these properties should enable the microparticles to mix with the blood and access the bleeding site without requiring mechanical intervention or auxiliary tools (Fig. 1c and Supplementary Movie 1). Subsequently, the OPA groups on the microparticle surface covalently crosslink with amines present on proteins in the imbibed blood and the tissue matrix, ultimately forming a mechanically strengthened and adhesive clot that halts hemorrhage and helps prevent rebleeding (Fig. 1d).

### Imbibition and crosslinking behaviors of BICMs

To investigate the potential imbibition and crosslinking behaviors of BICMs, we initially incubated a single oHA or xHA microparticle with rhodamine-labeled bovine serum albumin (Rho-BSA). Upon contact with the liquid, the oHA microparticle imbibed the liquid and swelled in diameter by ~30% within 5 seconds. Simultaneously, Rho-BSA concentrated on the oHA microparticle surface as a fluorescence halo (Fig. 1e). This behavior was similar to that observed with the xHA microparticle (Supplementary Fig. 6a). We reasoned that the inherent imbibition, driven by water absorption, facilitates the directional movement of Rho-BSA molecules toward the oHA or xHA microparticle surface. This automatic action also enables the microparticles to mix effectively with the liquid. After thorough washing, a fluorescence halo remained on the oHA microparticle surface, while no

detectable fluorescence was observed around the xHA microparticle. After blocking the OPA groups on oHA microparticles with ethanolamine, the resulting microparticles (EtoHA) exhibited limited crosslinking with proteins and susceptibility to fragmentation (Supplementary Fig. 6b). These results collectively suggest that proteins are crosslinked to the oHA microparticle surface through covalent bonding rather than physical interactions (e.g., hydrogen bonding or π-π stacking). We further characterized the oHA microparticles reacted with BSA using X-ray photoelectron spectroscopy (XPS), and Fourier transform infrared (FTIR) spectroscopy, confirming the formation of covalent bonds between OPA groups and amines on the oHA microparticle surface (Supplementary Fig. 6c and d).

In practical applications, BICMs function as a mass of oHA microparticles rather than individual units, working collectively to imbibe and mix with liquid. Thus, in addition to water absorption, the interstitial microchannels between oHA microparticles and their volume determine the liquid imbibition behavior through capillary suction. To investigate the influence of microchannels on liquid imbibition, we varied the pre-crosslinking density and microparticle size. We measured the weight gain of microparticles upon exposure to PBS for 10 seconds (defined as the imbibition rate) and after saturation (defined as the imbibition ratio). By adjusting the concentration of BDDE crosslinkers, we found that a higher pre-crosslinking density resulted in a faster imbibition rate of oHA microparticles (Fig. 1f). Note that, after exposure to PBS, highly pre-crosslinked oHA microparticles (13.5% and 16.5%) retained structural support and maintained microchannels for capillary suction, unlike those with low pre-crosslinked density (0 and 9.8%, Fig. 1f left). However, further increases in pre-crosslinking density reduced the imbibition capability of oHA microparticles after saturation, as indicated by a lower imbibition ratio, consistent with previous literature[39].

By decreasing the size of oHA microparticles (13.5% pre-crosslinking density) through adjustments in preparation conditions (Supplementary Fig. 1b and c), we observed a faster imbibition rate (Fig. 1g), consistent with the increased capillary rise height associated with smaller microchannel diameter[40]. However, further reduction in microparticle size resulted in a limited enhancement in the imbibition rate, alongside a decline in the imbibition ratio. Therefore, for this study, we adopted a pre-crosslinking density of 13.5% and a diameter of 75–120 μm for oHA microparticles to achieve both rapid and high imbibition.

### Grafting OPA groups onto microparticles accelerates clot formation and strengthens clots through covalent crosslinking

To investigate whether oHA microparticles rapidly form mechanically strengthened clots, we initially assessed their clotting performance by measuring the gelation time and modulus of the resulting clots using rheological analysis (Fig. 2a–c). Natural blood gelled within $117.1 \pm 12.5$ s, with a modulus of $0.18 \pm 0.03$ kPa (Fig. 2b). In contrast, the formation of clots comprising oHA microparticles and blood (denoted as oHA@Blood clots) occurred with a short gelation time ($2.1 \pm 0.2$ s), and these had a high modulus ($23.79 \pm 3.72$ kPa). The acceleration of clot formation and the increase in clot modulus were presumed to be a joint effect of introducing pre-crosslinked hyaluronic acid networks and the OPA groups presenting on microparticles. Indeed, in the absence of OPA groups, xHA@Blood clots formed in a longer time ($15.9 \pm 2.6$ s) with a lower modulus of $3.87 \pm 0.51$ kPa. To further investigate the effect of OPA groups on clot formation, we varied oHA microparticles with different densities of OPA groups (Fig. 2c). The results revealed that reducing or deactivating the OPA groups prolonged the gelation time and decreased the clot modulus, suggesting that the OPA group functions directly in accelerating clot formation and strengthening clots. More practically, we poured blood and oHA microparticles into a mold, ultimately forming a bulk oHA@Blood clot without any mechanical intervention

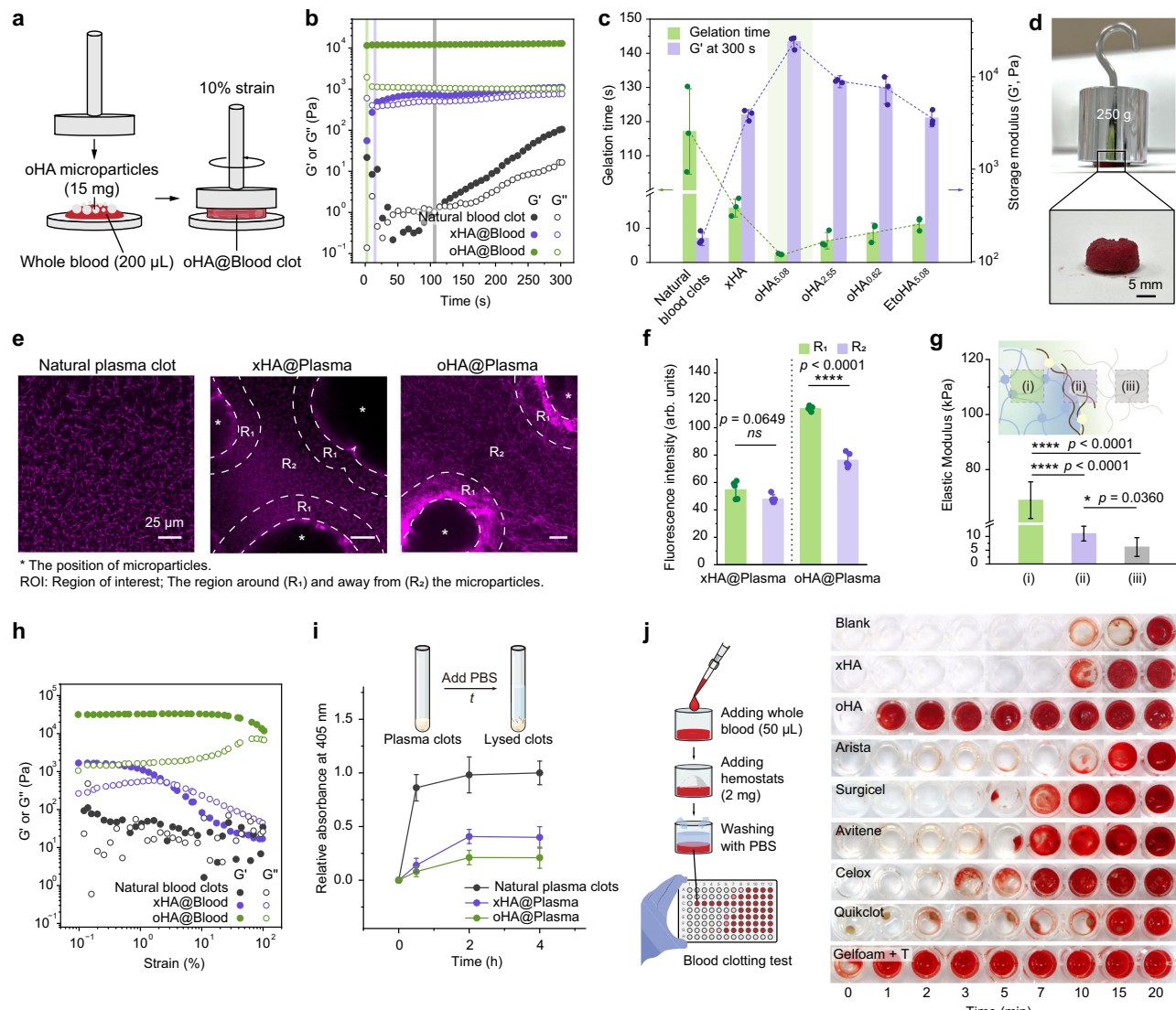

**Fig. 2 | Grafting OPA groups onto microparticles accelerates clot formation and strengthens clots through covalent crosslinking. a–c** Rheological analysis of blood clots with or without microparticles. Schematic illustration (**a**) and representative time sweeps (**b**) of the storage modulus (G′) and loss modulus (G″) for whole blood (200 μL), and whole blood (200 μL) with xHA (without OPA groups) or oHA (with OPA groups) microparticles (15 mg). The gelation time was defined as the time point at which G′ surpass G″. **c** Gelation time and G′ (measured via rheological tests) for blood clots with or without microparticles ($n = 3$ independent samples). The G′ was recorded at 300 s. **d** Digital photographs showing an oHA@Blood clot withstanding a weight of 250 g. The composite (oHA@Blood clot) was fabricated by adding 30 mg of oHA microparticles to 400 μL of whole blood within a mold (1.5 mm-diameter) and incubating at 37 °C for 300 s. **e–g** Identification of the oHA-clot interface. **e** Representative confocal laser scanning microscope (CLSM) images of a natural plasma clot, an xHA@Plasma clot, and an oHA@Plasma clot. Purple

fluorescence corresponds to the rhodamine-labeled plasma proteins. **f** Average fluorescence intensity of regions of interest (ROI, $R_1$ and $R_2$) in (**e**, $n = 5$ independent samples). **g** Average Young's modulus of three characteristic regions in an oHA@Plasma clot, assessed by AFM tapping mode ($n = 5$ independent samples). **h** Time-dependent plots for G′ and G″ versus strain rate for a natural blood clot, an xHA@Blood clot, and an oHA@Blood clot. **i** Fibrinolysis and anti-fibrinolysis behaviors of natural blood clots, xHA@Plasma clots, and oHA@Plasma clots, quantified by measuring the absorbance of supernatant at 405 nm ($n = 5$ independent samples). **j** Schematic illustration and dynamic clotting tests of blood with or without xHA microparticles, oHA microparticles, or commercially available hemostats ($n = 3$ independent samples). Data are presented as means ± s.d., and all statistical analyses were performed using an unpaired two-tailed Student $t$ test. * $p < 0.05$, ** $p < 0.01$, *** $p < 0.001$, **** $p < 0.0001$. Source data are provided as a Source Data file.

or auxiliary tools. The resulting clot was uniform, indicating a thorough mixing of oHA microparticles with blood. Furthermore, the clot withstood a weight of 250 g without being squashed (~ 31.2 kPa, Fig. 2d), exceeding the strength of natural clots (~ 2.2 kPa)[41].

We, therefore, investigated the enhancement in clot strength by examining the microstructure of clots. We used decellularized and fluorescent plasma to prevent the limitations in visual transparency. The oHA@Plasma clot was highly heterogeneous, containing regions of dense fluorescent plasma observed in conjugation with oHA

microparticles (Fig. 2e and Supplementary Fig. 7a). In contrast, the fluorescence was evenly distributed in natural plasma clots and xHA@Plasma clots (except at the locations of xHA microparticles). Quantitative analysis of fluorescence intensity revealed the formation of a highly crosslinked oHA-clot interface (Fig. 2f). Next, we evaluated the modulus of the oHA-clot interface using atomic force microscopy (AFM) tapping tests, a widely used method for characterizing biointerfaces[42]. The interface exhibited a higher modulus ($11.03 \pm 2.69$ kPa) compared to regions away from the microparticles

(6.15 ± 3.40 kPa, Fig. 2g), indicating a highly crosslinked and strengthened oHA-clot interface that fortified the composite clot.

To investigate whether the covalent crosslinking with blood proteins via oHA microparticles promotes clot stability (i.e., resistance to mechanical stress and fibrinolytic dissolution)[43], we initially carried out strain-sweeping measurements to simulate mechanical blood flow (Fig. 2h). The oHA@Blood clot exhibited no yielding during strain scanning (over 100%) and no disintegration as a cohesive entirety in PBS (Supplementary Fig. 7b). In contrast, the natural blood clot showed high sensitivity to strain switching, characterized by fluctuating modulus (G′ and G″), while the xHA@Blood clot exhibited a yield strain of ~ 2%. Therefore, larger strain damaged their structural integrity (Supplementary Fig. 7b). Next, we incubated plasma with tissue plasminogen activator (tPA) to evaluate the clot stability under physiological fibrinolysis (Fig. 2i). Samples were diluted with PBS at predetermined time points to track the relative absorbance at 405 nm ($A_{405}$) of the supernatant, which increased as the clot was lysed. The xHA@Plasma and oHA@Plasma clots exhibited lower $A_{405}$ values compared to natural plasma clots, indicating that the incorporation of polysaccharide polymers resists fibrinolysis. The $A_{405}$ of the oHA@Plasma clot was lower than that of the xHA@Plasma clot. Overall, these results demonstrate that by forming the highly crosslinked oHA-clot interface, oHA microparticles rapidly form mechanically strengthened clots with high resistance to mechanical stress and fibrinolysis, which is consistent with the benefits of FXIIIa-mediated covalent crosslinking within natural clots[5].

Lastly, we evaluated the in vitro blood clotting abilities of oHA microparticles compared with commercially available hemostats (Fig. 2j and Supplementary Fig. 7c). During the tests[26], uncoagulated and weak clots were washed away at predetermined time points until the supernatant became colorless (Fig. 2j left). The time to form a clot was ~ 19 min, whereas after adding oHA microparticles, the clotting time significantly decreased to ~ 1 min. This reduction is comparable to the clotting time enabled by a high concentration of thrombin, such as in the case of Gelfoam (an absorbable gelatin sponge) saturated with $300 \, \mu \, mL^{-1}$ topical thrombin (denoted as Gelfoam + T). Notably, this clotting time achieved with oHA microparticles was significantly shorter than that of xHA microparticles, Arista (absorbable polysaccharide particles), Surgicel (fibrillar oxidized cellulose), Avitene (microfibrillar collagen), Celox (chitosan granules), and Quikclot (zeolite granules). These results demonstrate that oHA microparticles accelerate clot formation and strengthen clots through covalent crosslinking before natural coagulation operates.

## Tissue adhesion of fortified clots

In addition to clot strength, clot adhesion equally contributes to hemostasis and the mitigation of rebleeding. Given that FXIIIa mediates covalent crosslinking both within clots and between clots and the tissue matrix, thereby enhancing clot adhesion, we next investigated the tissue adhesion performance of oHA@Blood clots during their in situ formation using a simplified model. Specifically, we applied drops of a Rho-BSA solution ($50 \, mg \, mL^{-1}$, PBS, pH = 7.4) onto an indocyanine green (ICG)-labeled gelatin matrix to simulate a tissue surface covered with blood. We then applied fluorescein isothiocyanate (FITC)-labeled oHA or xHA microparticles onto the matrix and rinsed them with PBS after 5 minutes (Fig. 3a). 3D confocal laser scanning microscope (CLSM) images revealed that oHA microparticles imbibed the interfacial Rho-BSA solution and crosslinked with Rho-BSA within the interstitial microchannels. After PBS rinsing, oHA microparticles remained firmly adhered to the gelatin matrix, whereas xHA microparticles were washed away (Supplementary Fig. 8).

To quantitatively evaluate the adhesion performance of the formed clots, we employed the standard lap shear and burst pressure tests (Fig. 3b and c). Compared to natural blood clots and clots formed using other materials, the oHA@Blood clot adhered to porcine skin demonstrated the highest shear strength (8.53 ± 0.67 kPa) and burst pressure (266.46 ± 51.04 mmHg). We further performed a hydraulic pressure test to assess the adhesion performance under conditions mimicking the continuous bloodstream[44]. The oHA@Blood clot adhered to the porcine skin withstood a hydraulic pressure of 117.22 ± 26.41 kPa, which was 2-fold higher than that of natural blood clots (Fig. 3d and Supplementary Movie 2). However, the addition of other materials resulted in a decrease in the tolerable hydraulic pressure compared to natural blood clots. We attribute the superior tissue adhesion of the oHA@Blood clot to covalent crosslinking with the tissue matrix, a mechanism that has been extensively studied[45]. Regarding adhesion, we also evaluated the adhesion of an oHA@Blood clot on a fresh, punched porcine aorta. The adhered oHA@Blood clot withstood continuous fluid flow (up to $180 \, mL \, min^{-1}$) repeated mechanical actions (~ 100 times) of bending, constricting, and twisting (Fig. 3e and Supplementary Movie 3).

We next evaluated the in vivo adhesion performance of oHA@Blood clots using a rat liver laceration model[46], and applied ICG-labeled oHA and xHA microparticles on the injury site, respectively. Excess materials were removed by flushing with saline, and the retention of microparticles was observed by in vivo imaging (Fig. 3f). The oHA microparticles displayed obvious fluorescence in situ up to two weeks. In contrast, the fluorescence of xHA microparticles was at a non-detectable level within three days post-implantation. The enhanced mechanical strength and tissue adhesion of the fortified clots prevented their detachment from the bleeding site, indicating a reduced risk of microparticle embolization (Supplementary Fig. 9). All the above results demonstrated that oHA@Blood clots had enhanced and stabilized tissue adhesion that resisted the pressure, mechanical actions, and the in vivo environment via covalent crosslinking with the tissue matrix as the function of FXIIIa[7].

## Biocompatibility and biodegradability

The biocompatibility and biodegradation of oHA microparticles were evaluated and compared with xHA microparticles, which are widely employed as soft tissue fillers with validated biocompatibility and biodegradability[36]. Residual BDDE crosslinker was thoroughly rinsed, in accordance with the FDA approval standards (<2 ppm, Supplementary Fig. 10). In addition, oHA microparticles crosslinked with blood proteins exhibited limited exothermic phenomena, with a lower temperature compared to Quikclot (Supplementary Fig. 11). Both LIVE/DEAD staining (Fig. 4a) and cell counting kit-8 (CCK-8) assays (Supplementary Fig. 12) revealed that oHA microparticles showed no obvious cytotoxicity (>90% cell viability) toward mouse fibroblasts, comparable to xHA microparticles. The hemolysis rate (Fig. 4b) of oHA microparticles was within the safety limit of 5% for hemostatic biomaterials[47]. We further evaluated the in vivo biocompatibility and biodegradability of oHA microparticles by dorsal subcutaneous implantation in rats, followed by assessment from 1 day to 3 months (Fig. 4c). Histological assessments from hematoxylin-eosin (H&E) staining revealed that oHA and xHA microparticles elicited comparable inflammatory responses, with slightly acute inflammation observed on day 1, transitioning to mild inflammation by day 7 post-implantation (Fig. 4d). Immunofluorescence intensity analysis demonstrated that oHA microparticles induced no significant differences in biomarkers related to inflammatory and foreign-body reactions compared with xHA microparticles and the sham group (Supplementary Fig. 13). Notably, both oHA and xHA microparticles exhibited gradual biodegradation, characterized by a comparable and progressive reduction in volume over the 3-month implantation period (Fig. 4e and Supplementary Fig. 14), avoiding the need for secondary removal[48]. These results indicate that the introduction of reactive OPA groups exhibits no obvious effect on the biocompatibility and degradability of xHA microparticles.

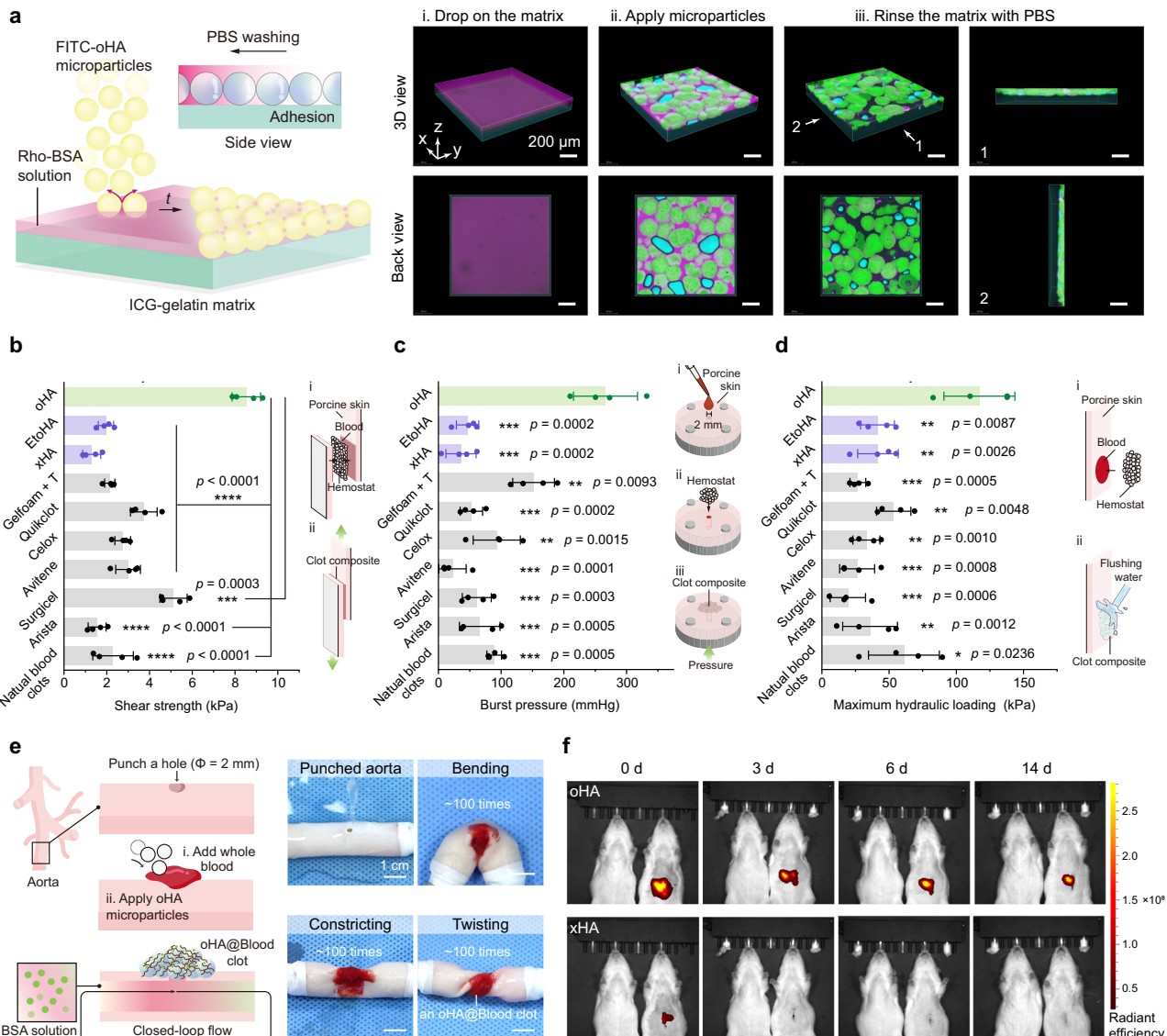

**Fig. 3 | Tissue adhesion of oHA@Blood clots. a** Schematic illustration and 3D CLSM images showing the adhesion behavior of oHA microparticles onto a gelatin matrix covered with a Rho-BSA solution (50 mg mL$^{-1}$ in PBS, pH = 7.4). **b–d** Quantification of the tissue adhesion. Lap shear tests (**b**), burst pressure tests (**c**), and tolerant hydraulic pressure tests (**d**) of natural blood clots, as well as blood clots formed using xHA, EtoHA, and oHA microparticles, and commercially available hemostats. Tests were performed after incubating the clots on fresh porcine skin for 5 minutes (*n* = 4 independent samples in **b–d**). **e** Adhesion of an oHA@Blood clot on a fresh, punched porcine aorta, demonstrating its ability to withstood

continuous fluid flow. **f** In vivo imaging system (IVIS) fluorescence analysis of the same rats examined on days 0, 3, 6, and 14 after applying oHA or xHA microparticles to a liver wound. The rat on the left serves as a negative control (i.e., no wound induction). Data are presented as means ± s.d., and all statistical analyses were performed using an unpaired two-tailed Student *t* test. * *p* < 0.05, ** *p* < 0.01, *** *p* < 0.001, **** *p* < 0.0001. The comparison was performed between each of the other hemostats versus oHA microparticles. Source data are provided as a Source Data file.

## Rapid hemostasis of liver injury in rat and coagulopathic pig models

After confirming their capacity to fortify clots, augment tissue adhesion, ensure biosafety, and facilitate biodegradability, we evaluated the hemostatic efficacy of oHA microparticles as BICMs in vivo. We induced models of in vivo rat and coagulopathic porcine liver injuries. We initially employed a rat liver volumetric defect model (2 mm in diameter and 3 mm in depth) created using a biopsy punch (Fig. 5a). BICMs, Arista, Surgicel, Quikclot, and Gelfoam + T (all weighing 0.02 g) were applied onto the injury sites, with cotton gauze used as a control (Fig. 5b). To quantitatively evaluate the hemostatic performance, we measured the time to hemostasis (the time with no blood leakage, Fig. 5c) and the blood loss until hemostasis (blood oozed from the

materials was collected using dry, pre-weighed gauze, Fig. 5d). BICMs showed a hemostasis time of 17.5 ± 6.1 s and a blood loss of 35.0 ± 30.4 mg. Conversely, Arista, Surgicel, and Quikclot exhibited longer hemostasis times (45.0 ± 21.2 s, 37.5 ± 28.1 s, and 65.0 ± 34.1 s, respectively) and greater blood loss (176.8 ± 134.6 mg, 96.2 ± 82.1 mg, and 300.2 ± 203.3 mg, respectively). Despite the rapid coagulation found in the above clotting tests, the hemostatic time (145.0 ± 50.8 s) and blood loss (450.7 ± 186.8 mg) of Gelfoam + T were considerably higher compared to BICMs, attributable to the inadequate tissue adhesion of the formed clots and the loss of thrombin during blood flow.

To assess potential inflammation and organ-specific responses, the injured liver tissues and blood samples were collected from the

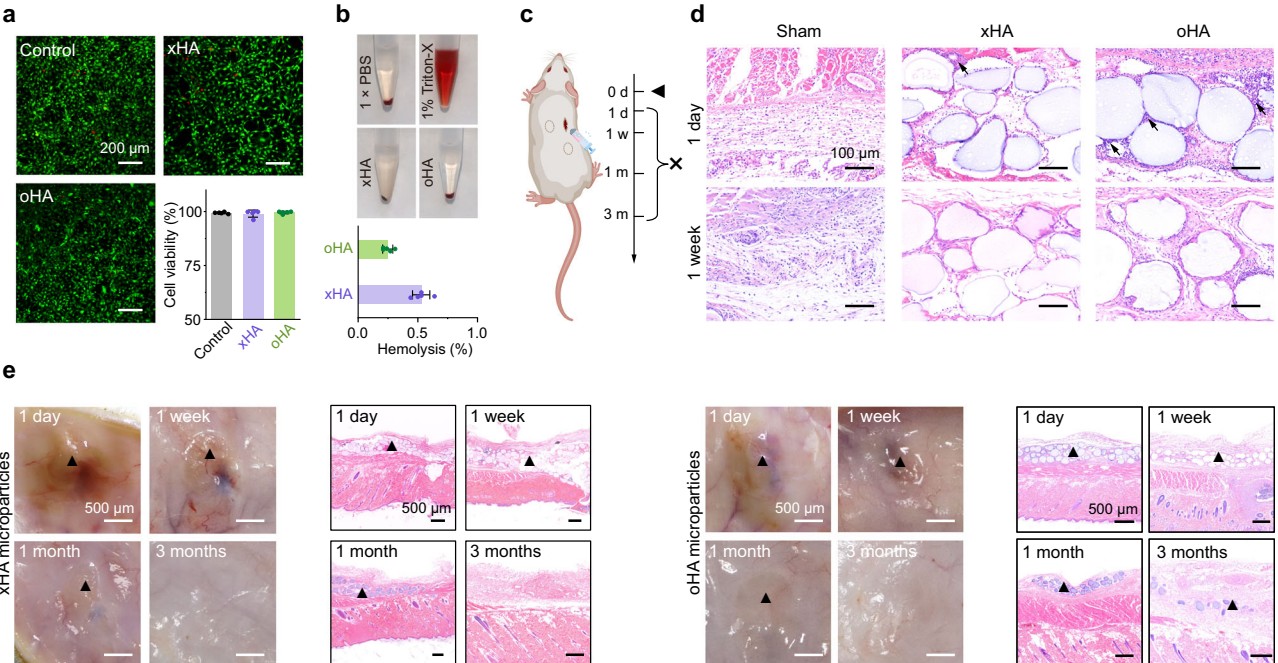

**Fig. 4 | In vitro and in vivo biocompatibility of oHA microparticles. a** In vitro cell viability of mouse fibroblast (L929) cells assessed using a LIVE/DEAD assay after 24 h of co-incubation with or without 20 mg of xHA or oHA microparticles (*n* = 5 independent samples). **b** Hemolysis rates of xHA and oHA microparticles, along with representative digital images (*n* = 5 independent samples). **c** Timeline of the subcutaneous implantation experiment for evaluating in vivo biocompatibility and biodegradation. Created in BioRender. Chen, T. (2025) https://BioRender.com/ x73e121. **d** Representative histology images with H&E staining on days 1 and 7 post-subcutaneous implantation. Black arrows indicate the infiltration of inflammatory cells. **e** Representative digital photographs and histology images with H&E staining of the subcutaneous tissue from the oHA and xHA microparticles-implanted groups at 1 day, 1 week, 1 month, and 3 months post-implantation. Black triangles indicate residual microparticles. Data are presented as means ± s.d. Source data are provided as a Source Data file.

animals at week one post-injury. Histological analysis revealed that BICMs induced a thin fibrotic capsule, comparable to that of Arista and Gelfoam + T, and thinner than that of Surgicel and Quikclot (Fig. 5e and Supplementary Fig. 15). The complete blood count and blood chemistry analysis (Supplementary Fig. 16) revealed no significant differences in inflammation-related blood cells or markers for organ-specific diseases among rats in all groups. These results indicate that BICMs did not impair the wound-healing process.

We then induced a coagulopathic porcine liver injury (Fig. 5f). Briefly, the coagulopathic status was induced by hemodilution, involving the exchange of 40% of the estimated total blood volume with three times the volume of cold 0.9% saline. This approach was confirmed based on the total protein (TP) levels and the maximum amplitude (MA) value measured by thromboelastography (TEG, Supplementary Fig. 17). After the hemodilution procedure, TP and MA decreased, indicating the successful establishment of the coagulopathic model. We created biopsy injuries (10 mm in diameter and 10 mm in depth) on the livers (four injuries per pig, for a total of twelve injuries), and compared the performance of BICMs with Surgicel, a time-tested and widely used hemostat.

BICMs were administered without the need for dressing or manual compression, showing a significantly decreased time to hemostasis (within 30 s) and reduced blood loss (1.00 ± 0.41 g) until hemostasis, compared to Surgicel (232.5 ± 78.9 s and 7.60 ± 1.42 g, Fig. 5g and h). For Surgicel, despite its brief application (~3 s) with gentle compression to position the dressing, blood leaked through the entire hemostat, and the masses of erythrocytes accumulated in the loose gaps between the hemostat and the tissue (Fig. 5i and j), indicating incomplete hemostasis. Note that no leakage of blood was observed on the surface after applying BICMs (Fig. 5i), and BICMs formed a tight seal with the deep injury tissue (Fig. 5j), indicating that a strengthened clot seamlessly conforms to the irregular tissue surface, effectively

preventing bleeding. The combined results demonstrated that BICMs halted hemorrhage rapidly in both normal and coagulopathic conditions.

## Prevention of rebleeding in a rabbit model of femoral artery injury and fluid resuscitation

We next employed a rabbit model of femoral artery injury and fluid resuscitation (Fig. 6a) to evaluate the capacity of BICMs to prevent rebleeding, which can occur when clots disrupt due to factors such as elevated blood pressure (BP) exceeding the mechanical and structural stability of the clot[23]. To induce this model, a 5-mm incision was made on the femoral artery, which was clamped proximally and distally to prevent bleeding. Prior to releasing the clamps to initiate bleeding, we standardized BP (below 40 mmHg) in all animals by performing a catheter hemorrhage from the carotid artery (Fig. 6b). We applied the hemostats (i.e., BICMs, Arista, Surgicel, Quikclot, and Gelfoam + T) onto the wound; a control group received no hemostat. After achieving initial hemostasis (*t* = 0), saline (0.9%) was infused via intravenous injection to raise and maintain BP above 60 mmHg. Every 5 minutes, we measured the blood loss by collecting the blood leaking from the wound, and a rebleeding event was defined as blood loss >0.1 g within a 5 minute interval. The rebleeding incidence (%) was defined as the ratio of the number of rebleeding events to the total number of assessment periods.

In the BICMs group, the clots formed at the wound effectively precluded rebleeding (Supplementary Movie 4), and the BP of the animals increased steadily with fluid resuscitation, stabilizing around 70 mmHg (Supplementary Fig. 18a). In contrast, the other four hemostat groups exhibited large hematomas beneath the primary clots, and rebleeding events were frequent (Supplementary Fig. 18b and Supplementary Movie 5). Specifically, rebleeding occurred only twice (4%) in the BICMs group. In the other groups, rebleeding

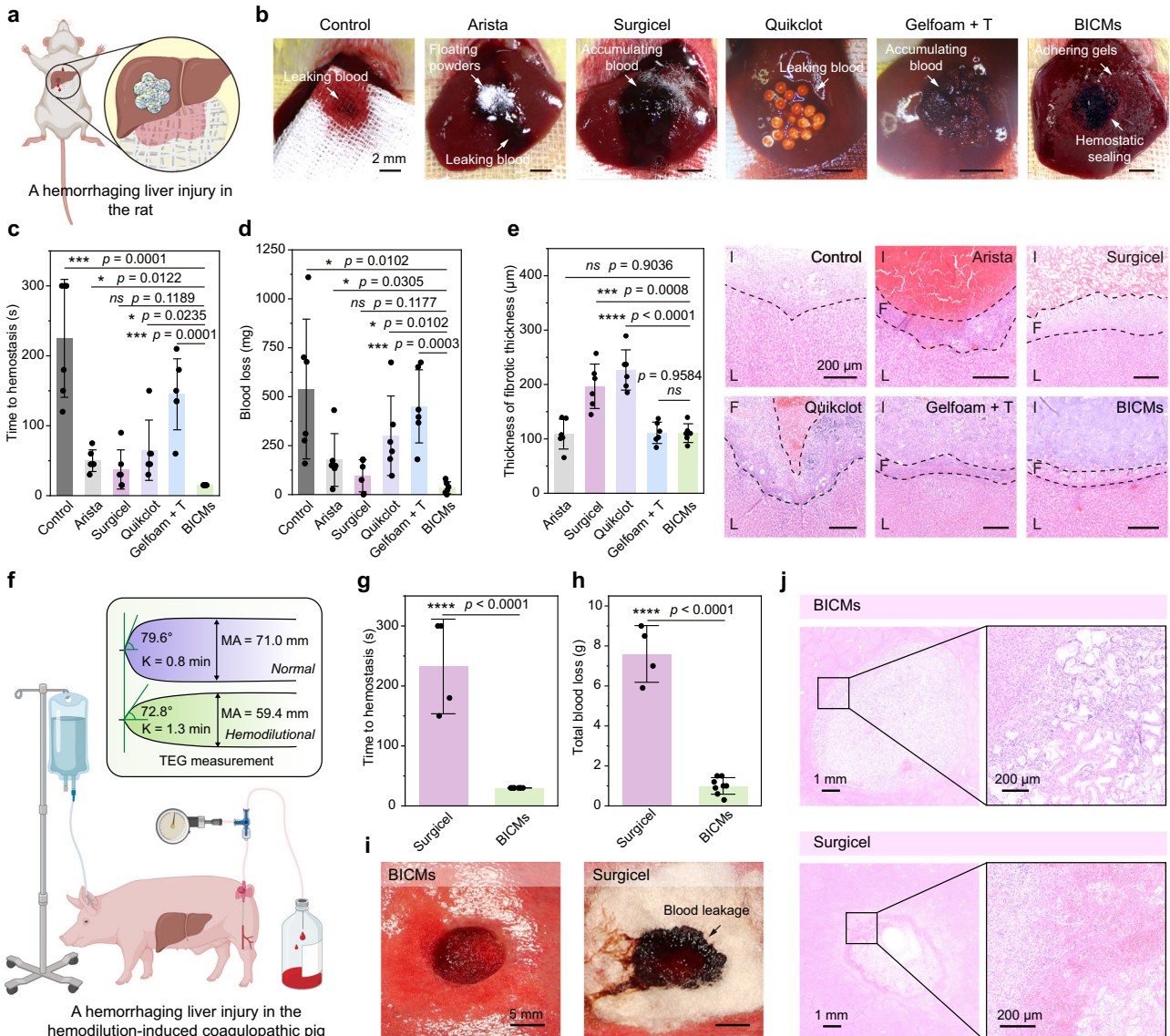

**Fig. 5 | In vivo hemostasis of hemorrhaging liver injury in rats and hemodilution-induced coagulopathic pigs. a** Schematic illustration of hemostasis using oHA microparticles in a rat liver injury model. Created in BioRender. Chen, T. (2025) https://BioRender.com/s64m575. **b–d** Hemostatic performance of gauze, Arista, Surgicel, Quikclot, Gelfoam + T, and BICMs. Representative photographs (**b**), time to hemostasis (**c**), and blood loss (**d**) during liver bleeding or hemostasis. **e** Representative histological images with H&E staining of injured livers, and the thickness of the fibrotic capsule 1 week post-injury. Dashed lines indicate the boundaries between the liver tissue (L), the fibrotic capsule (F), and the injury site (I). **f** Schematic illustration of the hemorrhaging liver injury model in hemodilution-induced coagulopathic pigs. Created in BioRender. Chen, T. (2025) https://BioRender.com/w64h091. **g, h** Time to hemostasis (**g**) and blood loss (**h**) during hemostasis in the BICMs and Surgicel groups (12 injuries in three pigs, $n = 8$ for BICMs and $n = 4$ for Surgicel). Blood loss was determined as the weight gain of gauze covering the hemostats. **i** Representative photographs during liver bleeding or hemostasis. **j** Representative histology images with H&E staining of injured livers 1-day post-injury. Data are presented as means ± s.d., and all statistical analyses were performed using an unpaired two-tailed Student $t$ test. * $p < 0.05$, ** $p < 0.01$, *** $p < 0.001$, **** $p < 0.0001$. Source data are provided as a Source Data file.

incidences were as follows: 51% in the control (natural blood clot), 42% in the Arista group, 38% in the Surgicel group, 34% in the Quikclot group, and 32% in the Gelfoam + T groups (Supplementary Fig. 18c and Fig. 6c). The cumulative blood loss was significantly reduced in the BICMs group (5.86 ± 4.92 g) compared to the control (43.66 ± 27.59 g), Arista (33.90 ± 28.75 g), Surgicel (27.90 ± 11.36 g), Quikclot (18.31 ± 5.55 g), and Gelfoam + T (19.53 ± 9.93 g) groups (Fig. 6d). Although no significant difference in survival was observed between the BICMs group and the other groups, a general trend toward improved survival was observed with the use of BICMs (Fig. 6e). One hour post-injury, we collected the injured femoral arteries and rinsed them with saline: the BICMs clot was still adhered to the vessel wall (Fig. 6f and g), whereas the clots in the other groups had completely

detached during collection (Supplementary Fig. 19). These results collectively demonstrate that BICMs form clots capable of withstanding elevated blood pressure and preventing rebleeding.

## Discussion

Inspired by the empirical evidence that FXIIIa-mediated covalent crosslinking within clots and between clots and the tissue matrix stabilizes clots[5,7], we here developed BICMs as a hemostat. The pre-crosslinked hyaluronic acid microparticles facilitate thorough mixing with blood via automatic imbibition; the externally grafted OPA groups enable covalent crosslinking with blood proteins and the tissue matrix. Upon contact with blood, BICMs form a fortified clot in situ immediately, exhibiting enhanced mechanical strength and tissue

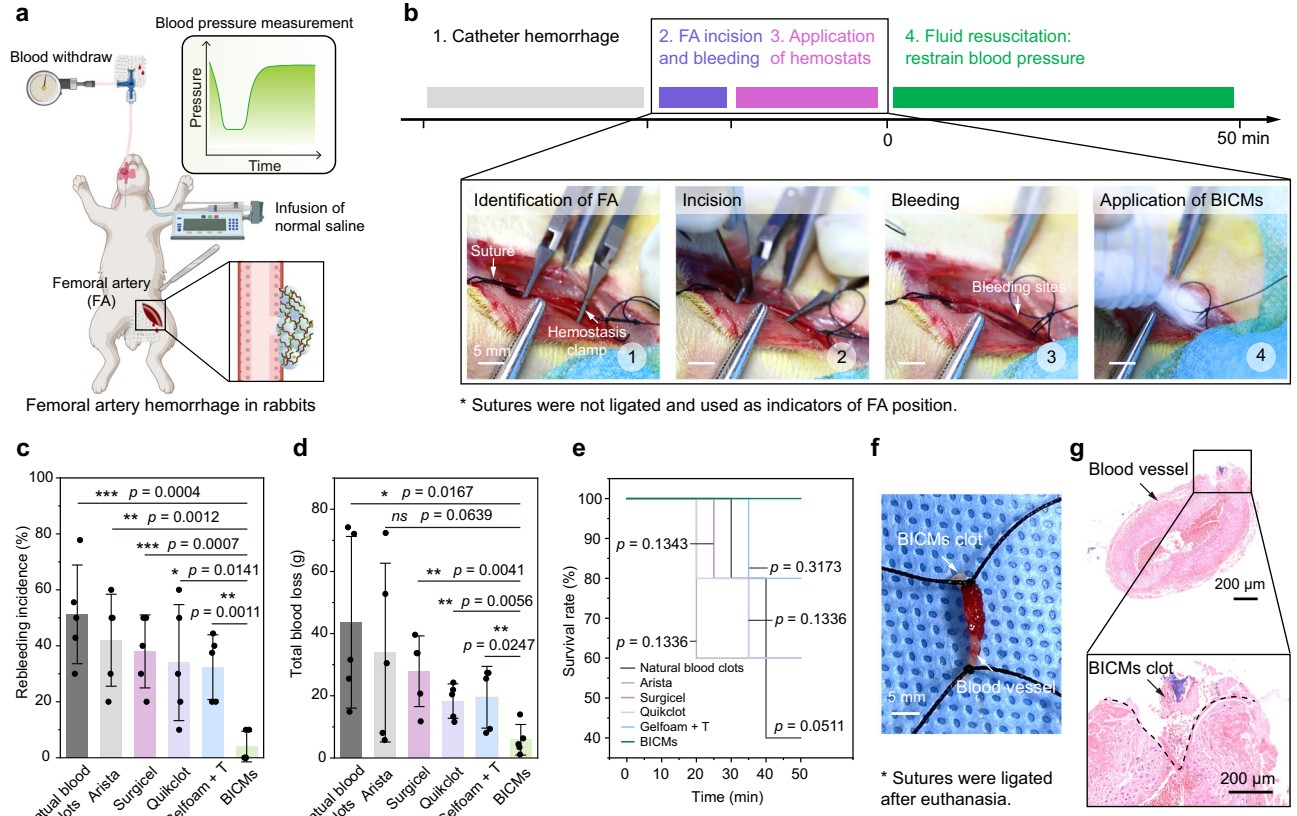

**Fig. 6 | In vivo hemostasis in a rabbit model of femoral artery injury and fluid resuscitation. a**, **b** Schematic illustration (**a**) and workflow diagram (**b**) for the model of femoral artery injury and fluid resuscitation. The illustration was created in BioRender. Chen, T. (2025) https://BioRender.com/y12r125. **c** Incidence of rebleeding during fluid resuscitation (*n* = 5 independent samples). **d** Total blood loss during initial hemostasis and fluid resuscitation (*n* = 5 independent samples). **e** Survival curves for rabbits during initial hemostasis and fluid resuscitation (*n* = 5

independent samples). **f**, **g** Representative photographs (**f**) and histological images with H&E staining (**g**) of a blood vessel after hemostasis using BICMs. Data are presented as means ± s.d. Statistical analyses were performed using an unpaired two-tailed Student *t* test in (**c**) and (**d**) and a log-rank Mantel-Cox test in (**e**). * *p* < 0.05, ** *p* < 0.01, *** *p* < 0.001, **** *p* < 0.0001. Source data are provided as a Source Data file.

adhesion. This performance is distinct from that of FDA-approved topical hemostats, which are designed to mimic, amplify, and/or leverage coagulation but fail to stabilize clots in the first place, as covalent crosslinking with these agents occurs only during the final phase of coagulation. BICMs achieved enhanced hemostatic performance in experimental models, including rats and coagulopathic pigs, characterized by significantly decreased hemostasis time and reduced blood loss. Moreover, BICMs effectively prevented blood-pressure-elevation-induced rebleeding in a rabbit model of femoral artery injury and fluid resuscitation. Notably, despite their prolonged degradation, BICMs did not impede tissue healing or cellular infiltration.

The design of BICMs confers potential feasibility in clinical utility. The single-component microparticle design facilitates the easy application of BICMs as a topical intervention and potentially via endoscope[49], without the need to mix multiple reactive components. Blood, which is traditionally considered to hamper the adhesion of sealants, in our case, serves as a reactive, structural component for BICMs; accordingly, there is no need for auxiliary pressure to repel blood. Another essential consideration is the use of OPA groups, which exhibit high reactivity toward amines yet maintain high stability in aqueous solution at room temperature. These properties differentiate BICMs from the currently available hemostats or sealants that require special storage conditions (e.g., dry, low temperature, and in vacuum)[45], further benefiting the clinical translation of BICMs.

Despite the promising features of BICMs, additional challenges regarding the clinical translation of the BICMs must be acknowledged

and overcome. Firstly, although we performed subcutaneous implantation experiments in rats, we did not compare the influence of microparticle geometries (i.e., microparticles produced by batch emulsion versus fragmentation methods). For example, irregular shape and non-uniform size might give rise to unwelcome aggregation and unpredictable embolization[50], and the geometry of implanted materials could influence the host foreign-body response[51]. While this experimental setting did not exhibit cases of significant thrombosis caused by BICMs, further investigation is required to determine the full extent of this phenomenon. Secondly, further investigations into the efficacy and biosafety of BICMs would be required before undertaking any clinical evaluations. Overall, we envision that BICMs will provide an effective tool for rapid hemostasis and prevention of rebleeding, promoting the development of hemostats.

## Methods

### Ethical Statement
The experimental operations in this study strictly complied in accordance with the appropriate ethical guidelines. All animal experiments were approved by the Institutional Animal Care and Use Committee (IACUC) at Shanghai Jiao Tong University (the approval number: A2023170).

### Materials
α-*N*-(9-Fluorenylmethyloxycarbonyl)-lysine (Fmoc-Lys-OH), *o*-Phthalaldehyde (Compound 1), N-Acetoxysuccinimide (Compound 2),

Hydroxy-PEG2-C2-PFP ester (Compound 3), Benzaldehyde (Compound 4), Heptaldehyde (Compound 5), and 3,4-dihydroxyphenylacetic acid (Compound 6) were purchased from Sigma-Aldrich and stored at suggested conditions. Hyaluronic acid (Mw: 1.2 MDa) was purchased from Bloomage Biotech Co., Ltd. (China). N,N-Dimethylformamide (DMF), dichloromethane (DCM), tetrahydrofuran (THF), triethylamine (TEA), methanol (MeOH), dimethyl sulfoxide (DMSO), ethanol, lithium tetrafluoroborate (LiBF$_4$), trimethoxymethane (TMO), and acetone were purchased from Titan Co. Ltd. (Shanghai, China). Bovine serum albumin (BSA, 66 kDa), rhodamine B isothiocyanate (RBITC), 1,5,7-triazabicyclo [4.4.0] dec-5-ene (TBD), 1-hydroxybenzotriazole (HOBT), hexamethylenediamine (HMDA), 1,4-Butanediol diglycidyl ether (BDDE), formylhydrazine, lead tetraacetate, paraformaldehyde (PFA), indocyanine Green amine (ICG-NH$_2$), 4-(4,6-Dimethoxy-1,3,5-triazin-2-yl)-4-methylmorpholin-4-ium chloride (DMTMM), nicotinamide, hyaluronidase (from bovine tests) and 4-Morpholineethanesulfonic acid (MES) were purchased from Sigma-Aldrich. Fluorescein Isothiocyanate-polyethylene glycol-NH$_2$ (FITC-PEG-NH$_2$, average Mw: 550 Da) was purchased from Leyan (China). Fibrinogen (from human plasma) was purchased from Sigma-Aldrich. Thrombin (from bovine plasma) was purchased from Beyotime (China). Recombinant human tissue plasminogen activator protein (tPA, active, Ab92637) was purchased from Abcam. All the other reagents are of analytical grade and are used without further purification.

### Monitoring reactions between compounds 1–6 and primary amines on lysine side chains

Fmoc-Lys-OH was dissolved in a trace of DMSO and diluted with PBS (1% DMSO, pH = 7.4) with final reaction concentrations of 400 μM (4.0 equivalents) or 110 μM (1.1 equivalents). Compounds 1–6 were dissolved in PBS and added into the above solution, respectively, with a final reaction concentration of 100 μM (1.0 equivalent). The reaction mixtures were stirred at room temperature (rt). At predetermined time points (0, 20, 40, 60, 80, 120, and 200 min), the conversion was monitored and calculated with high-performance liquid chromatography (HPLC), based on the consumption of Fmoc-Lys-OH ($n = 3$ independent samples). For the stability test, Compounds 1, 2 or 3 was dissolved in PBS (pH = 7.4), and stored at rt (800 μM) for defined periods of time (30 days for OPA, and 3 hours for NHS and PFP esters, respectively). Fmoc-Lys-OH was dissolved as described above with a final reaction concentration of 400 μM (4.0 equivalents). Compounds 1–3 were added into the above solution, respectively, with a final reaction concentration of 100 μM (1.0 equivalent) and reacted for 200 minutes at rt. The conversion at 200 minutes was recorded and compared ($n = 3$ independent samples). The HPLC apparatus was a Waters e2695 separations module, equipped with a Waters 2489 UV/vis detector. HPLC was performed using the following specifications: mobile phase, 40% acetonitrile / 60% water / 0.1% trifluoroacetic acid (TFA); column, XBridge C18 5 μm, 4.6 × 250 mm; flow rate, 1 mL min$^{-1}$; injection volume, 10 μL; column temperature, 25 °C; and the detect wavelength, 280 nm.

### Synthesis

The synthesis of methyl 3-(1,3-dimethoxy-1,3-dihydroisobenzofuran-5-yl)propanoate (OPA-Me)[37] was carried out as illustrated in Supplementary Fig. 3a. To introduce the amine group for modification, N-(6-aminohexyl)-3-(1,3-dimethoxy-1,3-dihydroisobenzofuran-5-yl)propenamide (OPA-NH$_2$) was synthesized. OPA-Me (0.5 g, 1.9 mmol) was dissolved in 2.5 mL of DMF. TBD (26 mg, 0.19 mmol) was added, followed by the addition of HMDA (870 mg, 7.49 mmol) to the mixture. The mixture was stirred for 24 h at room temperature. After the reaction was completed, the solvent was evaporated under vacuum, and the residue was collected and purified using a silica gel column chromatography with a mobile phase of DCM/MeOH/TEA (20:1:0.1

to give a white solid (yield: 80%), and characterized by $^1$H Nuclear magnetic resonance (NMR) and $^{13}$C NMR (Supplementary Fig. 3b and c).

### Preparation of xHA microparticles

HA (10 g) was added in batches into 100 mL of 0.25 M sodium hydroxide (NaOH) and dissolved. Then, BDDE was added to the HA solution and stirred for 10 minutes. The resulting viscous aqueous solution was poured dropwise into excess stirring castor oil (1 L). The mixture was homogenized for 5 min. The milky emulsion was further stirred at 800 rpm and 40 °C. After 48 h of reaction, excess ethanol was poured into the above emulsion to separate the oil and water phases. The upper oil layer was removed, and the residual aqueous phase was poured into a threefold volume of ethanol to precipitate white solids. The precipitants were redispersed into 0.9% saline (500 mL) and reprecipitated with ethanol three times. The final precipitants were dehydrated in ethanol and then dried under vacuum to obtain xHA microparticles. By changing the addition of BDDE (1.3, 1.5, 2.0 g), a series of microparticles with different pre-crosslinking densities were obtained. All microparticles were degraded to a solution with sulfuric acid before $^1$H NMR analysis. The pre-crosslinking density was defined as the percentage of BDDE per disaccharide unit. The resonance of ethyl protons (4H) of BDDE appeared at δ ~ 1.2, and the resonance of methyl protons of HA appeared at δ ~ 1.6. The pre-crosslinking densities of microparticles were determined to be 9.8%, 13.5% (selected for this study), and 16.5%, respectively. The original HA was considered as microparticles with 0% pre-crosslinking density. By changing the homogenization parameters (800, 4000, 8000, and 18000 rpm), microparticles of different sizes were obtained. The size distribution was defined using screen meshes: 45–80, 80–120, 120–200 (selected for this study), and 200–400 mesh.

Mechanical fragmentation is another manufacturing method that generates a large volume of microparticles rapidly in a single process, suitable for further in vivo experiments. Briefly, 10 g of HA was added into 100 mL of 0.25 M NaOH in batches and dissolved. Then, 1.5 g of BDDE was added to the HA solution and stirred for 10 min. The mixture was stewed at 40 °C for 8 h to form a hydrogel. Next, the bulk hydrogel was cut into smaller pieces, which then swelled in 1 L of 0.9% saline for equilibrium and mechanically forced through fine steel meshes to produce hydrogel microparticles with defined sizes. The hydrogel microparticles were dehydrated with ethanol to obtain precipitates, which were then redispersed in 0.9% saline (1 L) and reprecipitated using ethanol (3 L) for three times. The final precipitates were dehydrated in ethanol, dried under vacuum, and sieved to obtain xHA microparticles with a defined size.

### Preparation of oHA microparticles

xHA microparticles (10 g) were redispersed and swollen in 200 mL of MES buffer (pH = 5.2). OPA-NH$_2$ (0.7 g) was dissolved in 5 mL of DMSO and added to the mixture. Then DMTMM (5 g) was added in three equal portions at 8-hour intervals. The reaction mixture was stirred at 200 rpm and 37 °C for 48 h and monitored by HPLC. The microparticles were washed with 0.9% saline until there was no obvious UV/vis absorbance in the supernatant. Then, the obtained microparticles were redispersed in 1% TFA solution and stirred for 2 h to deprotect OPA functional groups. The microparticles were washed again with 0.9% saline, dehydrated using a series of gradient ethanol solutions, and then dried under vacuum to obtain oHA microparticles. The final product was stored under dry conditions at room temperature until use. By changing the reaction time (4, 12, 48 h), a series of microparticles with different OPA substitution degrees were obtained. The OPA substitution degree on oHA microparticles was quantified by a fluorescence differential method (described below) and was determined to be 0.62%, 2.55%, and 5.08% (selected for this study), respectively.

## Determination of BDDE residues via fluorospectrophotometry

BDDE residues can be evaluated via fluorescence intensity detection of the product formed from the reaction between BDDE and nicotinamide[52], exhibiting strong fluorescence at excitation/emission wavelengths of 388 nm/448 nm. Briefly, 20 μL of BDDE solution (8, 4, 2, 1, 0.5, and 0.25 μg mL$^{-1}$) was mixed with 10 μL of nicotinamide solution (125 mM), and incubated at 37 °C for 2 h. Next, 100 μL of acetophenone/ethanol solution (15%, w/w) and 100 μL of potassium hydroxide solution were added, and the mixture was incubated on ice for 15 min. Then, 0.5 mL of formic acid was added to the mixture, and the mixture was incubated at 60 °C for 5 min. The fluorescence was measured using a microplate reader (BioTek Synergy H1, Agilent). A sample of 100 mg of oHA microparticles was degraded using 1 mL of hyaluronidase solution (100 U mL$^{-1}$). The concentration of BDDE in enzymatic hydrolysate was characterized using a calibration curve.

## Preparation of EtoHA microparticles

oHA microparticles (1 g) were redispersed and swollen in 50 mL of 1% ethanolamine solution (PBS, pH = 7.4) at 37 °C for 24 h to block OPA groups. The microparticles were thoroughly rinsed with 0.9% saline, dehydrated using ethanol, and then dried under vacuum to obtain EtoHA microparticles.

## Preparation of FITC-xHA and FITC-oHA microparticles

xHA microparticles (1 g) were redispersed and swelled in 50 mL of MES solution (pH = 5.2). FITC-PEG-NH$_2$ (5 mg) was dissolved in 2 mL of DMSO and added to the mixture. Then, DMTMM (0.5 g) was added in three equal portions at 8 h intervals. The reaction was stirred at 200 rpm and 37 °C for 24 h, and the supernatant was monitored by UV/vis spectrometry. The microparticles were washed with 0.9% saline until there was no obvious absorbance of supernatant, dehydrated with ethanol, and then dried under vacuum to obtain FITC-xHA microparticles.

FITC-oHA microparticles were synthesized by OPA-amine reaction: oHA microparticles (1 g) were dispersed in PBS (50 mL, pH = 7.4). Then, FITC-PEG-NH$_2$ (5 mg) was dissolved in PBS (2 mL) and added to the above mixture. After stirring for 24 h at 37 °C, the microparticles were collected, washed, and dried to obtain FITC-oHA microparticles.

## Synthesis of ICG-oHA, ICG-xHA microparticles, and ICG-gelatin matrix

ICG-xHA and ICG-oHA microparticles were synthesized using a procedure similar to that for FITC-xHA and FITC-oHA microparticles. In this case, ICG-NH$_2$ (5 mg) was dissolved in DMSO (2 mL) and then added to the reaction mixture as a replacement for FITC-PEG-NH$_2$. To prepare the ICG-gelatin matrix, ICG-NH$_2$ (1 mg) was dissolved in 0.1 mL of DMSO, and then added into 10 mL of gelatin solution (200 mg mL$^{-1}$, PBS, pH = 7.4, 10 mL) at 40 °C. The mixture was poured into a confocal dish (Nunc$^{TM}$ glass-based dish 150680) cooled down to fabricate an ICG-gelatin matrix with a thickness of approximately 50 μm.

## Synthesis of Rho-BSA and Rho-plasma

Rhodamine B isothiocyanate (RBITC, 2 mg) was dissolved in 0.1 mL of DMSO and then added into the 10 mL of BSA solution (200 mg mL$^{-1}$, PBS, pH = 7.4). The mixture was stirred at room temperature for 3 hours, and then purified via dialysis (MWCO 3500) against distilled water. The obtained solution was lyophilized to obtain a pink solid, which was stored at −20 °C until use. RBITC (2 mg) was dissolved in 0.1 mL of DMSO and then added into 10 mL of fibrinogen solution (40 mg mL$^{-1}$, PBS, pH = 7.4). The mixture was stirred at room temperature for 3 hours, and then purified via dialysis (MWCO 3500) against distilled water. The obtained solution was lyophilized and stored at −20 °C until use. Plasma was obtained as supernatant by centrifugation (200 × g for 11 minutes) of whole citrated blood.

Rhodamine-labeled fibrinogen was dissolved in plasma (with a concentration of 40 mg mL$^{-1}$) to get Rho-plasma.

## Quantification of OPA substitution degree

The OPA substitution degree was roughly determined by HPLC analysis during the reaction. HPLC was performed using the following specifications: mobile phase, 15% acetonitrile / 85% water / 0.1% trifluoroacetic acid; column, XBridge C18 5 μm, 4.6 × 250 mm; flow rate, 1 mL min$^{-1}$; injection volume, 10 μL; column temperature, 35 °C; and the detection wavelength, 210 nm.

To quantitatively measure the OPA substitution degree on oHA microparticles, we performed the fluorescence differential method. 2 mL of FITC-PEG-NH$_2$ solution (2 mg mL$^{-1}$) was prepared, and oHA microparticles (5 mg) were added. After vortexing and incubating at 37 °C for 24 h, the characteristic absorbance of the supernatant was measured at 450 nm using a UV-vis light spectrometer. The concentration of FITC-PEG-NH$_2$ was determined using a calibration curve, and the OPA substitution is related to the differential of concentration (i.e., the amount of FITC-PEG-NH$_2$ that has reacted).

## Preparation of commercially available hemostats

For Quikclot (Advanced Clotting Sponge), zeolite granules were obtained from a mesh sponge and stored in a strictly dry environment. For thrombin-saturated gelatin sponges (Gelfoam + T), a 300 U mL$^{-1}$ thrombin solution (in PBS, pH = 7.4) was prepared. Gelatin sponges were cut into 1 mm × 1 mm × 1 mm pieces using surgical scissors. Then, the pieces were immersed in the thrombin solution. The pieces were collected; the excess thrombin solution was gently squeezed out, and the remaining pieces were collected for immediate use. Other commercially available hemostatic materials were used and stored according to the manufacturer's instructions.

## Evaluation of the imbibition and crosslinking for a single oHA/xHA microparticle

The oHA and xHA microparticles were first suspended in ethanol and placed on poly-D-lysine-coated glass slides. Following solvent evaporation, the microparticles were adhered to the slide. The procedure started with identifying a single microparticle in a bright field. Then, 20 μL of Rho-BSA solution (50 mg mL$^{-1}$, pH = 7.4) was dispensed onto the microparticle. The process was recorded with an inverted CLSM (Leica STELLARIS8). After the microparticles was incubated for 5 s the free Rho-BSA was washed out by flushing PBS in situ five times.

## Characterization of imbibition capability

To measure the imbibition ratio of oHA microparticles, each sample (approximately 50 mg) was poured into cell strainers (40 μm) that were pre-weighed ($W_s$), clean, and dry. The initial weight ($W_0$) of each strainer containing the sample was recorded. Then, the dry strainer was suspended on the liquid surface, with only one surface in contact with the liquid, maintaining a consistent contact area. The weight ($W_t$) of each strainer containing the sample was recorded again. To measure the imbibition rate, each sample was exposed to PBS for 10 seconds, and the rate was defined as the weight gain ratio in the initial 10 seconds ($n = 3$ independent samples). To measure the imbibition ratio after saturation, each sample was contacted with PBS for 24 hours, and the rate was defined as the final weight gain ratio ($n = 3$ independent samples).

$$Imbibition\ rate/ratio = \frac{W_t - W_0}{W_0 - W_s} \qquad (1)$$

## Rheological study

All rheological experiments were performed on a HAAKE MARS III rheometer at 37 °C. Citrated blood was obtained from sheep and stored in the tube with sodium citrate (109 mmol L$^{-1}$, blood/sodium

citrate = 9/1, v/v). To measure the dynamic interaction with samples and blood, 100 µL of calcium chloride (CaCl$_2$, 0.2 M) was added into 1 mL of citrated blood, and inverted gently for mixing. 200 µL of the prepared blood was placed on the plate (20 mm in diameter) with or without adding different microparticles (15 mg). Time sweeps were performed at a 10% strain [Controlled Deformation (CD) mode], 1-Hz oscillatory frequency, and a 0.5 mm gap for 300 s ($n$ = 3 independent samples).

To identify the mechanical stability of the clots, a rheological strain sweeping test was performed. Firstly, 200 µL of the above blood was placed on the plate (20 mm in diameter) with or without the addition of xHA or oHA microparticles (15 mg). Then, the samples were covered by a rotor at 37 °C for 300 s. Strain sweeps were performed at a 1 Hz oscillatory frequency and shear strain (γ) ranging from 0.1% to 100%. Points were collected over the log-scale range of γ values. The yield strain was determined as the strain value when G″ surpassed G′ ($n$ = 3 independent samples).

### Confocal imaging of clots

Citrated plasma was obtained as supernatant by centrifugation (200 × $g$ for 11 min) of whole citrated blood. Rho-plasma (1 mL) with CaCl$_2$ (0.2 M, 100 µL) was added into clear tubes. A 200 µL aliquot of Rho-plasma was dropped on the adhesive glass slide to form plasma clots (5 minutes, 37 °C). For xHA@Plasma and oHA@Plasma clots, 15 mg of xHA or oHA microparticles were added to the plasma and incubated at 37 °C for 5 min. All samples were incubated in a humid atmosphere to prevent dehydration. After incubation, all samples were imaged using an inverted CLSM (Leica STELLARIS8) with a 40 × objective lens. Fluorescence intensity was quantified using ImageJ (v.1.53).

### AFM measurements

Plasma was obtained as the supernatant of whole citrated blood by centrifugation (200 × $g$ for 11 min), and mixed with 100 µL of CaCl$_2$ (0.2 M). Then, 100 µL of the mixture was placed on a glass mold (0.5 mm depth), and 5 mg of oHA microparticles were immediately added into the mold. The formed samples were incubated at 37 °C for 5 min. Before measurement, 0.9% saline was added dropwise onto the sample until the sample was immersed. Nanomechanical measurements on the sample were conducted using a FastScan Bio AFM (Bruker, USA) in ramp mode ($n$ = 5 independent samples). Nanomechanical measurements were performed using an MLCT-O10 E Triangular probe (Bruker, USA) with a deflection sensitivity of ~ 37.00 N m$^{-1}$, spring constant of ~ 0.1000 N m$^{-1}$, and tip radius of ~ 6000 nm. Ramp parameters: ramp size of 2000 µm, ramp rate of 1.00 Hz, forward velocity of 4.00 µm s$^{-1}$, reverse velocity of 4.00 µm s$^{-1}$.

### Fibrinolysis tests

Plasma was obtained as the supernatant of whole citrated blood by centrifugation (200 × $g$ for 11 min). A 200 µL aliquot of the plasma containing tissue plasminogen activator (tPA, 250 ng mL$^{-1}$, Abcam) was added into a clear tube. xHA or oHA microparticles (15 mg) were added into the above tubes instantly, and the clotting was activated by adding 10 µL of CaCl$_2$ (0.2 M). Samples were incubated at 37 °C. At predetermined time points (0.5, 1, 2 h), a 1 mL aliquot of PBS was added into each tube to wash the lysed clots. The absorbance of the supernatant at 405 nm (A$_{405}$) increased as clots were lysed. A control group was prepared where neither microparticles nor CaCl$_2$ were added. Instead, 10 µL of PBS was added as a substitute. The relative absorbance at 405 nm was calculated using the following equation ($n$ = 5 independent samples):

$$Relative\ absorbance\ at\ 405\ nm = \frac{A_{sample}}{A_{ctrl}} \tag{2}$$

where $A_{sample}$ and $A_{ctrl}$ represent the absorbance at 405 nm of supernatant from experimental and control groups, respectively.

### In vitro blood clotting tests

A dynamic blood clotting test was performed to evaluate the clotting capability of materials[26]. Firstly, 100 µL of CaCl$_2$ (0.2 M) was added into 1 mL of citrated blood, and inverted gently for mixing. Next, 2 mg of microparticles or commercial hemostats were added into a 96-well plate, with each well containing 50 µL of the prepared blood at 37 °C, without mixing or applying pressure. At each time point, each well was washed with PBS to remove uncoagulated blood components or incohesive clots. The clotting time was defined as the time point at which no blood was flushed out from clots ($n$ = 3 independent samples).

### Tissue adhesion tests

In the following tests, 100 µL of CaCl$_2$ (0.2 M) was added to 1 mL of citrated blood, and inverted gently for mixing.

The lap shear test was performed following the ASTM F2255 standard. Fresh porcine skin (25 × 25 mm) was adhered to glass slides (80 × 25 mm) using cyanoacrylate glue and kept moist. A 100 L aliquot of the prepared blood was dispersed on the surface of two pieces of skin. A total of 10 mg of material was initially administered to the skin, and a second piece was placed on the first piece of skin instantly and gently. For natural clots, only blood was uniformly dispersed on the skin. The specimens were kept at 37 °C in a humidified environment for 5 min. One side of the sample was fixed, and increasing pull strength was applied to the opposite side using an INSTRON 68SC-1 tensile testing machine. Thereafter, the shear strength was calculated according to the following equation ($n$ = 4 independent samples).

$$Shear\ strength = \frac{F_{max}}{A} \tag{3}$$

where, $F_{max}$ represents the maximum pulling stress and $A$ is the adhesive area.

The burst chamber was produced following the ASTM F2392 Standard. Fresh porcine skin (thickness: 4 mm) was rinsed and trimmed into pieces (40 mm × 40 mm). A 2 mm-diameter hole was created on the skin using a biopsy punch. Before applying hemostats, the prepared blood (100 µL) was added to the hole to simulate the interfacial blood, then different hemostats (5 mg) were applied onto the skin without mixing or applying pressure. The specimens were maintained at 37 °C in a humified environment for 5 min. During the test, a 50 mL syringe was employed to administer water into the chamber and breach the defect. A pressure gauge was connected to the feeding tube to monitor the peak pressure, which was defined as the burst pressure ($n$ = 4 independent samples).

The hydraulic pressure test was performed to evaluate the tissue adhesion of materials[44]. Fresh porcine skin was washed and trimmed into pieces (25 mm × 25 mm) and attached to glass slides. A 100 µL aliquot of the prepared blood was uniformly dispersed on the surface of porcine skin with the same area (diameter: 10 mm). Then, 5 mg of materials were evenly dropped into the blood without mixing or pressure. The specimens were kept at 37 °C inside a humified environment for 5 min. A hydraulic giant with tunable hydraulic pressure was applied to test the adhesive strength. The maximum hydraulic pressure was recorded by a hydraulic pressure gauge ($n$ = 4 independent samples).

A hole (3 mm diameter) was created on the porcine aorta using a biopsy punch. Before applying hemostats, a 100 µL aliquot of the prepared blood was filled in the aorta. Then, oHA microparticles were applied to the simulated bleeding sites, and excess microparticles were removed. The aorta was connected to a liquid bath (Rho-BSA,

50 mg mL$^{-1}$, PBS, pH = 7.4) and a pump via silicone tubes to sustain flow (up to 180 mL min$^{-1}$) to evaluate the robustness of adhesion.

## In vivo adhesion performance of *o*HA microparticles

Firstly, the Sprague-Dawley (SD) rats (12-week-old, weighing 225–275 g) were anesthetized using isoflurane (1–3% in oxygen) in an anesthetizing chamber. Abdominal hair was shaved, and abdominal skin was disinfected with iodophor. The left liver lobe was exposed and put on wet gauze. A 3.0 × 1.5 cm section of the liver edge was transected with scissors[46]. ICG-oHA microparticles and ICG-xHA microparticles were applied on the bleeding sites and then gently pressed until hemostasis (*n* = 3 independent samples). Excess microparticles were flushed away by saline (0.9%). After hemostasis, the abdominal incision was closed. Rats that did not undergo the liver-transection operation served as the control group. The microparticles on livers were imaged by an IVIS imaging system (PerkinElmer, USA) 0, 3, 6, and 14 days after the operation and quantified in terms of average radiant efficiency [(p/sec/cm²/sr)/(μW/cm²)]. Control and ICG-oHA microparticle-implanted rats were euthanized after 14 days. Their major organs were collected and subjected to ex vivo fluorescence detection. For further histological analysis, the organs were fixed in 4% PFA for 24 h. To avoid potential sex influence on the assessment of tissue adhesion, animals of the same sex were used in this experiment.

## In vitro biocompatibility evaluation

The cell compatibility of xHA or oHA microparticles was tested by co-culture systems with mouse fibroblast (L929, Cat No. FH0534, provided by Shanghai Fuheng Biotechnology Co., LTD) cells. Briefly, cells were cultured in 24-well plates, and xHA or oHA microparticles (10 mg) were added into the co-culture system by a transwell chamber (Corning polycarbonate membrane cell culture inserts, with 6.5 mm diameter and 0.4 μm pore size, *n* = 5 independent samples). The cell viability was determined by a LIVE/DEAD staining kit (Beyotime), visualized with a confocal laser scanning microscope, and calculated by counting the number of live (green fluorescence) and dead (red fluorescence) cells in ImageJ (v.1.53).

The cell viability was further evaluated using CCK-8 assays. xHA and oHA microparticles (0.1, 0.5, and 1 g) were immersed in 20 mL of Dulbecco's Modified Eagle Medium (DMEM) at 37 °C for 24 h to obtain the supernatant as leachates. L929 cells were cultured in 96-well plates, and the above experimental culture medium were added into wells. The control pristine DMEM, the DMEM with high-density polyethylene (HDPE) immersed (0.1 g mL$^{-1}$), and the DMEM with 5% DMSO were used as a blank, negative, and positive control, respectively. The cells were incubated at 37 °C for 24 h and 48 hs in 5% $CO_2$. The cell viability was measured using CCK-8 assays and calculated with the following equation (*n* = 6 independent samples):

$$Cell\ viability\,(\%) = \frac{A_{sample} - A_{blank}}{A_{ctrl} - A_{blank}} \times 100\% \qquad (4)$$

where $A_{sample}$, $A_{blank}$, and $A_{ctrl}$ represent the absorbance intensity at 450 nm of experimental/negative/positive groups, blank groups, and control groups, respectively.

To evaluate the hemocompatibility, erythrocytes were collected and incubated in different conditions at 37 °C for 24 h. xHA and oHA microparticles (1 g) were swollen for equilibrium, collected, and then immersed in an extra 10 mL of PBS at 37 °C for 24 h to get the supernatant as leachates. Triton X-100 (1%, v/v) and PBS (pH = 7.4) solution were used as positive and negative controls, respectively. The hemolysis phenomenon induced by leachates or control groups after incubation was determined by measuring the concentration of released hemoglobin collected by centrifuging (200 × *g*, 5 min). The hemolysis rate (%) of different groups was calculated as the following equation

(*n* = 5 independent samples):

$$Hemolysis\,(\%) = \frac{A_{sample} - A_{PBS}}{A_{Triton} - A_{PBS}} \times 100\% \qquad (5)$$

where $A_{sample}$, $A_{PBS}$, and $A_{Triton}$ represent the absorbance intensity of hemoglobin (410 nm) in the leach liquor, PBS, and Triton-X groups, respectively.

## In vivo biocompatibility and biodegradation tests

To evaluate the biocompatibility and biodegradability of oHA microparticles in vivo, dorsal subcutaneous implantation in a rat model was performed. Before implantation, all surgical instruments were sterilized. oHA and xHA microparticles were sterilized for 3 h under UV light. Male SD rats (12-week-old, weighing 225–275 g) were anesthetized using isoflurane (1–3% in oxygen) in an anesthetizing chamber. The back hair was shaved, and the back skin was disinfected with iodophor. Subcutaneous pockets were accessed by a 10-mm incision and widened by blunt dissection on both sides of the spine. Either 5 mg of oHA or xHA microparticles (*n* = 3 independent samples for each endpoint) were placed into the subcutaneous pockets, respectively. Rats with only a surgical incision were used as sham controls. Then, the incision was closed with interrupted sutures. At 1 day, 1 week, 2 weeks, 1 month, and 3 months post-implantation, the rats were euthanized via $CO_2$ inhalation. Partial subcutaneous tissues were excised and fixed in 4% PFA for 24 h for further histological and immunofluorescence analyses. The residual volume was determined by computing the product of the residual implant's cross-sectional area, illustrated in the digital image, and the implant's thickness, as determined in the H&E section findings. To avoid potential sex influence on the assessment of biocompatibility and biodegradation, animals of the same sex were used in this experiment.

## In vivo hemostatic performance in rat liver injury model

Firstly, male SD rats (12-week-old, weighing 225–275 g) were anesthetized using isoflurane (1–3% in oxygen) in an anesthetizing chamber. Abdominal hair was shaved, and abdominal skin was disinfected with iodophor. The left liver lobe was exposed, wetted with saline, and then placed on dry, weighed gauze. An injury (2 mm in diameter and 3 mm in depth) was made by a biopsy punch. Different materials (i.e., BICMs, Arista, Surgicel, Quikclot, and Gelfoam + T) of the same weight (0.02 g) were gently placed on the bleeding sites (*n* = 6 independent samples). For the control group, no hemostat except gauze was employed. The bleeding sites were observed, and the time to hemostasis was defined as the point when no blood was leaking. The blood oozing from the materials was collected using dry, pre-weighed gauze, and blood loss was characterized by the weight gain of gauze during hemostasis. After hemostasis, the abdominal incision was closed. One week after hemostasis, blood was collected for complete blood counting and blood chemistry (*n* = 3 independent samples), respectively. After euthanasia by $CO_2$ inhalation, injured livers were excised and fixed in 4% PFA for 24 h for further histological analysis. To avoid the potential sex influence on the assessment of hemostatic performance, animals of the same sex were used in this experiment.

## In vivo hemostatic performance on hemodilution-induced coagulopathic porcine liver

Domestic female pigs (3-month-old, weighing 40–50 kg) were anesthetized by injecting Zoletil 50 into the muscle at 4–6 mg per kilogram and endotracheally intubated. Anesthesia was maintained with 1–3% isoflurane. An intraarterial catheter was placed in the leg artery for pressure measurement, blood sampling, and withdrawal. Peripheral catheterization into the ear vein was performed for injection. The pigs were hemodiluted by exchanging 40% of the estimated total blood volume with threefold the volume of cold 0.9% saline (4 °C). The body

temperature was lowered to 35 °C and maintained during the experiment. Coagulopathic status was confirmed by the TP levels and the MA values of TEG.

A laparotomy was performed to expose the liver, and moist gauze was used to isolate the liver. A biopsy wound (10 mm in diameter and 10 mm in depth) was created on the liver by a standard biopsy punch. Then, the bleeding blood at the first minute was collected using dry, pre-weighed gauze to assess the bleeding rates. Next, BICMs (1 g) were applied without dressing and manual compression. For another group, Surgicel (Fibrillar Absorbable Hemostat, 1 g) was applied to the injured site over a short period (~3 s), with gentle compression to position in situ. No compression was used for subsequent observation. The bleeding status was assessed every 30 s. In the case of residual blood leaking, applying more hemostats is necessary until hemostasis. The time to hemostasis was recorded. Blood oozing from the materials was collected by dry, pre-weighed gauze, and blood loss was measured by the weight gain of pre-weighed gauze. Four injuries in each pig had no noticeable effect on invasive blood pressure and bleeding rates. The abdominal cavity was closed in a layered fashion, and the pigs were woken from general anesthesia and extubated. One day after initial hemostasis, the pigs were humanely euthanized, and the wound regions were excised and fixed in 4% PFA for 24 hours for histological analysis. To avoid potential sex influence on the assessment of hemostatic performance, animals of the same sex were used in this experiment.

### In vivo hemostatic performance in a rabbit femoral artery injury model

Male New Zealand White rabbits (6-month-old, weighing 3–3.2 kg) were anesthetized with Zoletil 50 and maintained with isoflurane (1–3% in oxygen) via a mask during the experiment. Before injury, the carotid artery and ear vein were catheterized to monitor BP and saline infusion, respectively. Then, the left femoral artery was exposed, isolated, and clamped. A 5 mm length incision was made on the artery, and sutures on two sides were used as indicators. Prior to releasing the clamps, blood was withdrawn from the carotid artery to decrease the BP below 40 mmHg. After removing the clamps, blood was lost from the femoral artery incision. For the natural clots group, the blood gushed out of the wound pocket, and pre-weighed gauze was used to press the injury site and remove excess blood outside the wound pockets until hemostasis. For the Arista, Surgicel, Quikclot, Gelfoam + T, and BICMs groups, each hemostat was applied to the bleeding site, and pre-weighed gauze was used to collect leaking blood. After initial hemostasis ($t = 0$), saline (0.9%) was infused at 3 mL kg$^{-1}$ min$^{-1}$ to raise and maintain BP above 60 mmHg. During resuscitation, BP was recorded every 5 min. Every 5 min, the blood loss was collected by dry, pre-weighed gauze. Experiments were stopped if rabbits died during resuscitation or if the time reached 50 min. Rabbits that did not survive the initial catheter hemorrhage were excluded from the analysis. After euthanasia by excessive intravenous injection of pentobarbital sodium, injured arteries were excised, washed with saline, and fixed in 4% PFA for 24 h for further histological analysis. To avoid potential sex influence on the assessment of hemostatic performance, animals of the same sex were used in this experiment.

### Histological analysis

Fixed tissue samples were dehydrated through gradient ethanol and embedded in paraffin wax blocks. Then, the paraffin sections were dewaxed in xylene, rehydrated with gradient ethanol, and washed with deionized water. Next, the sections were subjected to H&E staining. Subsequently, the sections were dehydrated and scanned.

### Immunofluorescence analysis

The expression of targeted proteins (i.e., α-SMA, CD3, TNF-α, CD68, CD163, and VEGF) was analyzed after immunofluorescence staining of the collected tissues. Paraffin sections were dewaxed, rehydrated, subjected to epitope retrieval, and blocked by incubation with BSA for 45 min at room temperature. The sections were covered with diluted primary antibodies and incubated for one hour at room temperature. The sections were washed with PBS, incubated with secondary antibodies for another hour, mounted with a mounting medium containing DAPI for 20 minutes, and then scanned. The primary antibodies included mouse anti-αSMA (ab7817, 1:200, Abcam), mouse anti-CD3 (ab5690, 1:200, Abcam), mouse anti-TNFα (ab1793, 1:200, Abcam), mouse anti-CD68 (ab201340, 1:200, Abcam), mouse anti-CD163 (ab156769, 1:200, Abcam) and rabbit anti-VEGF receptor 1 (ab32152, 1:200, Abcam). The secondary antibodies included donkey anti-rabbit IgG Alexa Fluor 488 (ab150073, 1:1000, Abcam) or goat anti-mouse IgG TRITC (ab6718, 1:1000, Abcam). The fluorescence intensity of the expressed antibodies was quantified via ImageJ (v.1.53) and calculated using normalized analysis. All analyses were blinded concerning the experimental conditions.

### Statistics and reproducibility

Experiments were performed with at least triplicate samples, and representative experiments (e.g., micrographs) were from at least three consistent repetitions. Statistical analysis was performed using MATLAB (v.R2018b) and Prism 10.0 (GraphPad). Data are presented as means ± s.d. For survival rate analysis, a log-rank Mantel-Cox test was used. In other statistical analyses between two data groups, a two-sided Student's $t$ test or a one-way ANOVA test was used.

### Reporting summary

Further information on research design is available in the Nature Portfolio Reporting Summary linked to this article.

## Data availability

The data supporting the findings of this study are available within the article and its supplementary files. Any additional requests for information can be directed to and will be fulfilled by, the corresponding authors. Source data are provided in this paper.

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

## Acknowledgements

This work was financially supported by the National Natural Science Foundation of China (32121005, L.Z., and 22475129, Q.L.), National Key Research and Development Program of China (2024YFA1107600, Y.Y.), 173 Plan Project (2019-JCJQ-ZD-359-00, L.Z.), and the Interdisciplinary Program of Shanghai Jiao Tong University (YG2022ZD009, W.G. and YG2024QNA19, Z.Y.).

## Author contributions

T.C., L.Z., and Q.L. were responsible for the experiment concept design, and data analysis. T.C., L.Z., and Q.L. wrote the manuscript. T.C., C.X., B.B., and Q.Z. prepared and characterized materials. T.C., C.X., and X.C. evaluated the biocompatibility and biodegradability of the materials and analyzed the data. T.C., J.Z., C.X., and Z.Y. performed the animal experiments and analyzed the data. T.C., X.H., and L.J. performed the fluorescence imaging and analyzed the data. L.Z., Q.L., W.G., Z.Y., and Y.Y. provided financial support. L.Z. and Q.L. supervised all aspects of this work. All authors commented on the manuscript and its revisions.

## Competing interests

The authors declare no competing interests.
