## [Transparent Peer Review file · Nature Communications]

Covalently reactive microparticles imbibe blood to form fortified clots for rapid hemostasis and prevention of rebleeding

Corresponding Author: Professor Linyong Zhu

Version 0:

Reviewer comments:

Reviewer #1

(Remarks to the Author)

This manuscript discusses a novel hemostatic product (blood-imbibing and crosslinking microparticles, BICMs) which can absorb blood into microparticles and crosslink themselves with blood proteins and the tissue matrix. Because this process does not require fibrin generation and FXIIIa, it can localize blood and stabilize a forming clot prior to fibrin crosslinking, which would theoretically improve hemostasis and prevent rebleeding. The efficacy of BICMs was assessed using various models of animal injuries, and BICMs were more effective at stopping bleeding, adhering to the wound, and stopping rebleeding compared to other commonly used hemostats. I believe that this manuscript is generally useful and advances the field of biomedical engineering for hemorrhage control; however, the authors are suggested to respond to the following points:

Major revisions:

1. Because Surgicel, Arista, Celox, and Avitene are chemically unreactive hemostats, there is concern in the efficacy comparison between these and the chemically reactive oHA particles. I recommend expanding comparisons to other more reactive products—such as QuikClot (which contains FXII-activating kaolin) and/or hemostatics containing thrombin—for Figures 2j, 5, and 6.
2. I recommend conducting a calorimetry experiment to determine whether oHA and xHA microparticles produce an exothermic reaction upon crosslinking protein, as heat generation has previously been an issue in hemostats such as QuikClot and caused many burn injuries to patients.

Minor revisions:

1. Line 68: Please discuss the molecular mechanism and enzymes which allow OPA groups to crosslink with amine groups on blood proteins without coagulation cascade activation?
2. Line 75: reactivity was assessed during 30 days of storage in PBS. Why is this relevant? Many hemostatic sealants are not stored in aqueous solution.
3. Line 82-84: “through capillary suction outside and water absorption into microparticles” is unclear what the authors are trying to say. Also, it is unclear how absorbing blood helps the microparticles in accessing the bleeding site (ie. the source) after being administered.
4. There is a bit of confusion in the difference between oHA microparticles and BICMs, as the terms seem to be used interchangeably and the difference is not clearly defined.
5. I recommend the authors clearly describe the commercial hemostats which were used in comparison throughout the study. For example, Surgicel comes in both powder (Surgicel Powder Absorbable Hemostat) and dressing (Original Absorbable Hemostat) forms, and it is unclear which was used. This is an issue throughout the paper for all hemostats used.
6. Line 122: These references provide the rate of blood flow in arteries without injury, but the rate of blood loss is much higher during injury as blood will be flowing freely under higher pressure. I recommend providing numbers which more accurately reflect the rate of blood flow in situations where BICMs are being used. Also, how does 10 grams compare to the mass of BICMs which were applied in animal models, or are expected to be applied in clinical situations?
7. In addition to the previous point: is 100 mL of blood in 10 seconds the maximum rate of imbibition, or is it the maximum

volume of blood that can be imbibed before saturation? The total volume of blood that is lost during massive hemorrhage can exceed litres, so I suggest clarifying this, as well as the implications following microparticle saturation.

8. Line 230: The exact methods for the liver injury model are not clearly described. Some concerns include: the way it is described in the methods sounds like a wet gauze placed under the liver was used to collect lost blood. Is this correct, and if so, why not dry gauze? For the control group, was the cotton gauze placed atop of the injury weighed, or the wet gauze underneath? Was surgical dressing used, and if so, why not Surgicel powder, which would be more comparable to Arista and the oHA microparticles?

9. Line 236: I recommend reporting the hemostasis times and blood loss of the other groups here, rather than stating it is "obviously superior to other groups".

10. In addition to the previous point: please clarify whether the absorbency/imbibing qualities of the hemostatic agents would significantly influence the perceived blood loss in this experiment. The blood absorbed by the hemostatic agents or dressing should still classify as total blood lost from the body in a surgical setting.

11. Line 244: Coagulopathic porcine liver injury model is not clearly described. Were the oHA microparticles applied alongside dressing? Was the Surgicel dressing or powder? Were commercial hemostats used as intended (manual compression, etc)?

12. Please comment on whether BICMs have the risk of embolization after application?

13. Methods section: please clarify the following:

a. Lines 328 - 330: Minor grammatical errors

b. Line 356: Citrated whole blood: where was the blood sourced and how was it collected?

c. Animals used: age and sex of animals?

d. Line 423: Why was a trans-well system used to evaluate in vitro biocompatibility rather than direct co-incubation of microparticles with cells?

e. Line 443: centrifugation speed should be reported in x g rather than rpm.

Reviewer #2

(Remarks to the Author)

The manuscript presents a covalently reactive hemostat made of blood-imbibing and crosslinking microparticles, engineered to rapidly form fortified clots with improved mechanical stability and tissue adhesion. The authors demonstrate how this hemostat engages with blood through imbibition, forming covalent bonds with blood proteins and the tissue matrix to facilitate prompt clot formation. Compared to commercially available hemostats, these microparticles, synthesized from pre-crosslinked hyaluronic acid (HA) using a 1,4-butanediol diglycidyl ether (BDDE) crosslinker and subsequently grafted with o-phthalaldehyde (OPA) groups, exhibit superior hemostatic efficiency, as shown by the reduced hemostasis time and blood loss in rat and coagulopathic pig models. Further in vivo studies in rabbits confirm the efficacy of these microparticles, showing effective prevention of rebleeding triggered by elevated blood pressure. This work presents a novel material that can potentially lead to the clinical advancement of hemostatic agents, offering robust and effective solutions for bleeding management. The study was generally well conducted, but there are several issues to be addressed for further improving the manuscript.

Major comments

1. In Extended Data Figure 2a, six different chemical compounds are presented; however, the manuscript discusses the conversion results for only three of these compounds. Given that all six compounds are shown, it would be expected that results for all compounds are provided or that the rationale for selecting only three is clarified.

2. The schematic illustration in Figure 1d emphasizes covalent bonding exclusively. Is it possible that the oHA microparticles interact with blood proteins and tissue matrix via non-covalent crosslinking such as hydrogen bonding and pi-pi stacking? Potential contribution of other types of chemical interactions to rapid hemostasis may need to be discussed for additional crosslinking mechanism.

3. The rationale behind the optimization of the oHA microparticles is vague. The author's selection of 100 μm diameter with a 13.5% pre-crosslinking density lacks justification, especially considering that other variables showed higher imbibition rates. Further explanation of the author's choice of these parameters is required.

4. Emphasizing and presenting additional data on tissue adhesion more thoroughly would strengthen the manuscript. While the hydraulic pressure test in Figure 3b provides valuable insight, additional experiments, such as a lap shear test, could further support the suggested advantages of the oHA microparticles in tissue adhesion.

5. In Figure 4a, it is recommended to quantify the LIVE/DEAD staining data to clarify the biocompatibility of the microparticles more effectively. In Figure 4e, in vivo degradation profile of the implanted oHA microparticles needs to be quantified and compared with that of xHA microparticles.

6. Although immunostaining for macrophages was performed in Extended Data Figure 10, the possibility of immunogenicity and inflammation should be more carefully investigated given the use of BDDE crosslinker.

7. The minor injury created in the pig model may limit the assessment of the hemostat's effectiveness, especially given the choice of a larger animal model intended to validate its function. Similarly, the injury applied in the rat model may also be too minimal to thoroughly evaluate the hemostatic properties. Larger injuries might provide a more rigorous test of the hemostat's performance and better support its potential clinical applicability.

8. The oHA microparticles outperformed control hemostats (Arista, Surgicel) in hemostasis tests in Figure 5 and 6, but I'm not sure whether they are the best controls to be compared with the oHA microparticles. Are there any rationales to choose such control hemostats for comparison?

Minor comments

1. On page 5, line 118, "Supplementary Information" needs to specify figure number.
2. On page 6, the reference to Figure 2c is missing in the text.
3. On page 6, line 143, the reference to Figure 3c is incorrect, which should be corrected to Figure 2d.
4. In the "Rapid hemostasis of liver injury in rat and coagulopathic pig models" section, the figures are incorrectly labeled as Figure 4b-d and 4f-h, which should be corrected to Figure 5b-d and 5f-h. Also, the reference to Figure 5a is missing.
5. While the time point in Extended Data Figure 2 is mentioned as 200 minutes, the manuscript text refers to "three hours," creating a discrepancy that should be corrected for consistency.

Version 1:

Reviewer comments:

Reviewer #1

(Remarks to the Author)

My previous concerns have been addressed. One minor question was why sheep blood was used for the rheology testing. Additionally, the text should be thoroughly cleaned up throughout before publishing. There are many minor english mistakes and areas for improvement, and clean-up by an experienced science editor is needed. For example, in the abstract, providing quantitative results rather than saying 'outstanding' would be better. Another example, is that fibrin is referred to as 'fibrins'. There are many other examples.

Reviewer #2

(Remarks to the Author)

The authors have addressed all issues raised by this reviewer. Accordingly, the issues are now mostly resolved, and the manuscript has been improved. I do not have any other comments to the authors.

Response to Reviewers' comments:

This response letter refers to comments on the original manuscript NCOMMS-24-69080-T.

We thank the Reviewers for their insightful comments, prompting us to enhance the manuscript. Addressing these comments has significantly improved the experimental evidence, supporting our study's findings.

5 In response to Reviewer #1's suggestion, we have expanded comparisons using Quikclot and thrombin-soaked Gelfoam in clotting tests, as well as *in vivo* rat and rabbit models. New experimental evidence substantiates the efficacy of BICMs in halting bleeding and preventing rebleeding. These data now well illustrate the hemostatic efficacy of our new BICMs hemostat. Given the insightful suggestions from Reviewer #1, we have added a detailed description of commercial controls, alongside the methods and results of our animal experiments. These revisions
10 promote the accuracy and readability of our updated manuscript.

Given the concerns of Reviewer #2 regarding the optimization and working principle of BICMs, we have included extended characterization and evaluation focused on the imbibition ratio and the impact of possible non-covalent interactions. According to the recommendations provided, we incorporated the burst pressure test and the lap shear test to support the tissue adhesion of BICMs. Moreover, the manuscript includes quantification of cell
15 viability by LIVE/DEAD staining, quantification of the *in vivo* degradation profile of BICMs, and an analysis of potential risk regarding BDDE crosslinker. These additions confirm the biocompatibility and biodegradability of BICMs.

We deeply appreciate the time and work of the Editor and Reviewers.

Reviewer #1

Comment:

5 This manuscript discusses a novel hemostatic product (blood-imbibing and crosslinking microparticles, BICMs) which can absorb blood into microparticles and crosslink themselves with blood proteins and the tissue matrix. Because this process does not require fibrin generation and FXIIIa, it can localize blood and stabilize a forming clot prior to fibrin crosslinking, which would theoretically improve hemostasis and prevent rebleeding. The efficacy of BICMs was assessed using various models of animal injuries, and BICMs were more effective at stopping bleeding, adhering to the wound, and stopping rebleeding compared to other commonly used hemostats. I believe
10 that this manuscript is generally useful and advances the field of biomedical engineering for hemorrhage control; however, the authors are suggested to respond to the following points:

Response:

We are very appreciative to the reviewer for recognizing the significance of our study to the field.

15

Major revisions:

1. Because Surgicel, Arista, Celox, and Avitene are chemically unreactive hemostats, there is concern in the efficacy comparison between these and the chemically reactive oHA particles. I recommend expanding comparisons to other more reactive products—such as QuikClot (which contains FXII-activating kaolin) and/or
20 hemostatics containing thrombin—for Figures 2j, 5, and 6.

Response:

Thank you for highlighting the issue, and we agree on the importance of using commercial controls. Based on these recommendations, we have expanded comparisons using Quikclot and thrombin-saturated Gelfoam
25 (Gelfoam + T), as both are reactive hemostatic products. We have made corresponding revisions to the manuscript, Figure 2j, Figure 3b-d, Figure 5b-e, and Figure 6c-e (along with the corresponding Supplementary Figures and methodologies). These revisions include a comparison of BICMs with the aforementioned products regarding *in vitro* clotting ability, tissue adhesion, *in vivo* hemostatic efficacy, and biosafety.

Corresponding revision 1:

Response Figure 1 | *In vitro* clotting evaluation. **a**, Schematic illustration and dynamic clotting tests of blood with or without xHA or oHA microparticles, or indicated commercially available hemostats. **b**, The blood clotting time of natural blood clots, and the blood incubated with xHA/oHA microparticles, and commercially available hemostats (i.e., Arista, Surgicel, Avitene, Celox, Quikclot, and Gelfoam + T).

5 We initially evaluated the *in vitro* blood clotting abilities of Quikclot and Gelfoam + T, and compared them to oHA microparticles (Responsive Figure 1). These details are now included in Figure 2j and Supplementary Fig. 7c, and the manuscript (Lines 187-193, highlighted in red), which now states:

10 “The time to form a clot was ~19 minutes, whereas after adding oHA microparticles, the clotting time dramatically decreased to ~1 minute, similar to the clotting time enabled by a high concentration of thrombin [i.e., Gelfoam (absorbable gelatin sponge) saturated with 300 U mL⁻¹ topical thrombin, abbreviated as Gelfoam + T]. Notably, this clotting time enabled by oHA microparticles is considerably reduced compared to xHA microparticles, Arista (absorbable polysaccharide particles), Surgicel (fibrillar oxidized cellulose), Avitene (microfibrillar collagen), Celox (chitosan granules), and Quikclot (zeolite granules).”

15 **Corresponding revision 2:**

Response Figure 2 | In vivo hemostatic performance of Quikclot, Gelfoam + T, and BICMs. Representative photos (a), time to hemostasis (b), and blood loss (c) during liver bleeding/hemostasis.

In the rat liver injury model, we evaluated the hemostatic performance of BICMs compared to Quikclot and Gelfoam +T by measuring the time to hemostasis and the level of blood loss (Response Figure 2). These details are included in Fig.5b-d and the manuscript (Lines 267-273, highlighted in red), which now states:

“BICMs showed a hemostasis time of 17.5 ± 6.1 s and a blood loss of 35.0 ± 30.4 mg. Conversely, Arista, Surgicel, and Quikclot exhibited prolonged hemostasis times (45.0 ± 21.2 s, 37.5 ± 28.1 s, and 65.0 ± 84.3 s) and substantial blood loss (176.8 ± 134.6 mg, 96.2 ± 82.1 mg, and 300.2 ± 203.3 mg). Despite the rapid coagulation found in the above clotting tests, the hemostatic time (145.0 ± 50.8 s) and the blood loss (50.7 ± 186.8 mg) of Gelfoam + T exhibited considerably increased values compared to BICMs, attributable to the inadequate tissue adhesion of formed clots and the elimination of some thrombin during blood flow.”

In addition, we evaluated the potential inflammation and organ-specific responses associated with hemostats employed in the liver injury model. These details are now presented in Fig.5e, Supplementary Figs.15 and 16, and the manuscript (Lines 275-280, highlighted in red), which now reads:

“Histological analysis revealed that BICMs induced a thin fibrotic capsule comparable to that of Arista and Gelfoam+T, and thinner than that of Surgicel and Quikclot (Fig.5e and Supplementary Fig.15). The complete blood count and blood chemistry analysis (Supplementary Fig.16) revealed that there was no significant difference in inflammation-related blood cells and markers for organ-specific diseases among rats in all groups.”

Corresponding revision 3:

Response Figure 3 | *In vivo* hemostasis in a rabbit model of femoral artery injury and fluid resuscitation. a. Representative photos of failed and successful hemostasis treated with Quikclot, Gelfoam + T and BICMs, respectively. **c.** Total blood loss during initial hemostasis and fluid resuscitation. **d.** Survival curves for rabbits during initial hemostasis and fluid resuscitation. Statistical analysis was conducted using an unpaired two-tailed Student *t*-test in (b) and (c), and using a log-rank Mantel-Cox test in (d). ns $p \geq 0.05$, * $p < 0.05$, ** $p < 0.01$, *** $p < 0.001$, **** $p < 0.0001$.

In the rabbit model of femoral artery injury and fluid resuscitation, we assessed the capacity of BICMs to prevent rebleeding after initial hemostasis when compared with Quikclot and Gelfoam +T (Response Figure 3). These details are included in Fig.6c-e, Supplementary Figs. 18 and 19, Supplementary Video 5 and the manuscript (Lines 318-323, highlighted in red), which now states:

“Specifically, rebleeding only occurred two times (4%) in the BICMs group, while the rebleeding incidence was 51%, 42%, 38%, 34%, and 32% in the control (natural blood clot), Arista, Surgicel, Quikclot, and Gelfoam+T groups, respectively (Supplementary Fig.18c and Fig.6c). The cumulative blood loss decreased from 43.66 ± 27.59 g (natural blood clot), 33.90 ± 28.75 g (Arista), 27.90 ± 11.36 g (Surgicel), 18.31 ± 5.55 g (Quikclot), and 19.53 ± 9.93 g (Gelfoam+T) to 5.86 ± 4.92 g (BICMs, Fig.6d).”

Collectively, BICMs exhibited improved hemostatic efficacy, as indicated by their reduced *in vivo* hemostasis time, lowered blood loss, and decreased rebleeding incidence compared to Quikclot and thrombin-mediated active hemostatic materials, reactive hemostatic products.

2. I recommend conducting a calorimetry experiment to determine whether oHA and xHA microparticles produce an exothermic reaction upon crosslinking protein, as heat generation has previously been an issue in hemostats such as QuikClot and caused many burn injuries to patients.

5 **Response:**

Thank you for highlighting the issue. To expand biosafety knowledge on oHA microparticles, we have investigated the potential exothermic effect of oHA microparticles compared to Quikclot Advanced Clotting Sponge (ACS). As demonstrated in Response Figure 4, unlike the violently exothermic Quikclot ACS (highest temperature up to 67.6 °C), oHA microparticles crosslinked with blood proteins exhibited negligible exothermic phenomena at a lower temperature. This demonstrates the safety of oHA microparticles without causing additional burn injuries to patients. These details are included in Supplementary Fig.11 and the manuscript (Lines 239-241, highlighted in red), which now reads:

“Additionally, oHA microparticles crosslinked with blood proteins exhibited limited exothermic phenomena with a lower temperature compared to Quikclot (Supplementary Fig.11).”

15

Response Figure 4 | Evaluation of an exothermic reaction. a, Digital photos (top) and infrared thermal images of oHA microparticles and Quikclot contacting blood. b, The curves of the highest temperature in the dish over time.

Minor revisions:

- 20 1. Line 68: Please discuss the molecular mechanism and enzymes which allow OPA groups to crosslink with amine groups on blood proteins without coagulation cascade activation?

Response:

Thank you for pointing out this issue. We have revised the manuscript (Lines 69-73, highlighted in red) to include a discussion of the molecular mechanism that allows OPA groups to crosslink with amine groups on blood proteins without coagulation cascade activation. It now reads:

5 *“To support covalent crosslinking with blood proteins independent of coagulation cascade activation (i.e., in the absence of enzymatic reactions or catalysts), we chose the abundant primary amine groups present on proteins as crosslinking targets. Reactions between o-phthalaldehyde (OPA) groups and primary amines are rapid and chemoselective for the formation of phthalimidine linkages under physiological conditions^{36,37}.”*

- 10 2. Line 75: reactivity was assessed during 30 days of storage in PBS. Why is this relevant? Many hemostatic sealants are not stored in aqueous solution.

Response:

Thank you for highlighting the issue.

- 15 Despite the prevalence of hemostatic materials, specifically those based on blood-concentrated hemostasis in a dry state, they are exposed to body fluids once used in situations involving bleeding. The objective of this experiment was to evaluate the effectiveness of reactive-groups-based hemostats in actual application scenarios. Thus, it is critical to determine whether their reactivities are compromised by the physiological aqueous environment. From a clinical translation perspective, the stability of reactive groups in an aqueous solution
20 facilitates manufacturing.

As illustrated in Supplementary Fig.2, the stability of the OPA groups in a physiological aqueous solution enables complete cross-linking with proteins. Nevertheless, the unavoidable hydrolysis of reactive esters in the same environment occurs alongside their crosslinking reactions, limiting the practical use of these moieties.

- 25 3. Line 82-84: “through capillary suction outside and water absorption into microparticles” is unclear what the authors are trying to say. Also, it is unclear how absorbing blood helps the microparticles in accessing the bleeding site (ie. the source) after being administered.

Response:

Based on this comment from the reviewer, we have revised the manuscript (Lines 87-90, highlighted in red) for a clear and accurate description. It now reads:

5 *“In principle, once administered, the oHA microparticles located surrounding the bleeding site should imbibe the nearby blood into their interstitial microchannels between microparticles through capillary suction. Thereafter, oHA microparticles retain water from the imbibed blood within the pre-crosslinked network of each microparticle.”*

To demonstrate this process, we utilized a transparent protein solution (BSA, 50 mg mL⁻¹, indicated using a yellow dye) to simulate the trajectory of oHA microparticles (Supplementary Video 1), *i.e.*, from above the liquid to the deep and narrow bottom (bleeding site).

10

4. There is a bit of confusion in the difference between oHA microparticles and BICMs, as the terms seem to be used interchangeably and the difference is not clearly defined.

Response:

15 Thank you for highlighting the issue, we apologize for the confusion over the use of these two terms. We have revised the manuscript (Lines 259-261, highlighted in red) to provide clear definitions that illustrate the use of these terms. It now reads:

“After conforming their capacity to fortify clots, augment tissue adhesion, ensure biosafety, and facilitate degradability, we used oHA microparticles as BICMs to examine the hemostatic efficacy in vivo.”

20 Additionally, a meticulous examination of the contexts where oHA microparticles and BICMs were employed at each point in the text was conducted, leading to linkages to minimize confusion arising from the use of these terms.

BICMs were defined as the microparticles that leverage the blood-imbibing and -crosslinking mechanism. Specifically, oHA microparticles are a specific formulation of BICMs composed of reactive OPA-modified exteriors and pre-crosslinked, absorbent hyaluronic acid networks. According to the design of BICMs, oHA microparticles were prepared and optimized. After confirming the capability of oHA microparticles to strengthen clots, and enhance tissue adhesion, biosafety, and biodegradability, we selected oHA microparticles as BICMs for further *in vivo* hemostasis evaluation.

25

5. I recommend the authors clearly describe the commercial hemostats which were used in comparison throughout the study. For example, Surgicel comes in both powder (Surgicel Powder Absorbable Hemostat)

30

and dressing (Original Absorbable Hemostat) forms, and it is unclear which was used. This is an issue throughout the paper for all hemostats used.

Response:

5 Thank you for bringing to our attention an important issue. We have incorporated a more detailed description of commercial hemostats where they initially occurred in the results section (Lines 187-193, highlighted in red). It now reads:

10 *“The time to form a clot was ~19 minutes, whereas after adding oHA microparticles, the clotting time dramatically decreased to ~1 minute, similar to the clotting time enabled by a high concentration of thrombin [i.e., Gelfoam (absorbable gelatin sponge) saturated with 300 U mL⁻¹ topical thrombin, abbreviated as Gelfoam + T]. Notably, this clotting time enabled by oHA microparticles is considerably reduced compared to xHA microparticles, Arista (absorbable polysaccharide particles), Surgicel (fibrillar oxidized cellulose), Avitene (microfibrillar collagen), Celox (chitosan granules), and Quikclot (zeolite granules).”*

15 The commercially available hemostatic materials employed in the manuscript are consistent with those outlined above.

6. Line 122: These references provide the rate of blood flow in arteries without injury, but the rate of blood loss is much higher during injury as blood will be flowing freely under higher pressure. I recommend providing numbers which more accurately reflect the rate of blood flow in situations where BICMs are being used. Also, how does 10 grams compare to the mass of BICMs which were applied in animal models, or are expected to be applied in clinical situations?

20

Response:

25 Thank you for this insightful comment. Notably, 10 g of oHA microparticle can rapidly imbibe over 100 mL of liquid within 10 seconds, exceeding the volume of blood ejected from the ventricles per cardiac cycle (70-80 mL per heartbeat, Chruscik, A. *et al.*). However, we recognized the original statement lacks a solid scientific foundation. As indicated by the reviewer’s concerns, a conspicuous incongruity emerges between the quality used in our animal experiments (0.02 to 1 g) and the quality demanded for actual clinical applications. Consequently, we removed the mention of 10 g from the original text.

30

References:

Chruscik, A., Kauter, K., Windus, L., Whiteside, E. & Dooley, L. *Fundamentals of Anatomy and Physiology*. (University of Southern Queensland, 2021).

- 5 7. In addition to the previous point: is 100 mL of blood in 10 seconds the maximum rate of imbibition, or is it the maximum volume of blood that can be imbibed before saturation? The total volume of blood that is lost during massive hemorrhage can exceed litres, so I suggest clarifying this, as well as the implications following microparticle saturation.

10 **Response:**

Thank you for pointing out the significant issue.

Regarding the concern of the maximum rate of imbibition and maximum volume of blood that can be imbibed before saturation: 100 mL of blood in 10 seconds is the maximum rate of imbibition rather than the maximum volume of blood that can be imbibed following saturation. In brief, microparticles initially imbibe blood into the interstitial channels between microparticles via capillary action, manifested as imbibition rate. Subsequently, the water present in the blood was absorbed from the interstitial channels into the interior of the hydrophilic pre-crosslinked network, facilitating more blood being imbibed, manifested as the imbibition ratio after saturation. To clarify this, we have extended the imbibition ratio of microparticles. As presented in Response Figure 5, the selected microparticles (10 g) can absorb approximately 150 mL of liquid when saturated.

20

Response Figure 5 | The effects of (a) pre-crosslinking density and (b) microparticle size on the imbibition rate (in 10 seconds) and imbibition ratio (after saturation) of oHA microparticles. All values were represented as means ± s.d. (n = 3). All statistical analyses were performed using an unpaired two-tailed Student *t*-test. * *p* < 0.05, ** *p* < 0.01, *** *p* < 0.001, **** *p* < 0.0001.

These details are now included in the manuscript (Lines 119-134, highlighted in red), which now reads:

“We measured the weight gain of microparticles upon exposure to PBS for 10 s (defined as imbibition rate) and following saturation (defined as imbibition ratio).”

5 “However, further increases in pre-crosslinking density result in the limited imbibition capability of oHA microparticles following saturation with reduced imbibition ratio, consistent with the findings reported in the literature³⁸.”

“However, further reducing the size of oHA microparticles caused a limited enhancement in imbibition speed, alongside a decline in the imbibition ratio. Therefore, in this study, we adopted a pre-crosslinking density of 13.5% and a diameter of 75~120 μm for oHA microparticles by considering the fast and high imbibition.”

10 **Regarding the implications following microparticle saturation:** Crosslinking with the blood proteins may occur before the microparticles reach saturation, consistent with *in vitro* clotting tests. Therefore, the oHA@Blood clot, which was effectively formed, robust, and adhesive, prevented further bleeding at the site of the hemorrhage. In the context of a formed oHA@Blood clot, the microparticles undergo a gradual absorption of fluid, reaching a state of saturation.

15

8. Line 230: The exact methods for the liver injury model are not clearly described. Some concerns include: the way it is described in the methods sounds like a wet gauze placed under the liver was used to collect lost blood. Is this correct, and if so, why not dry gauze? For the control group, was the cotton gauze placed atop of the injury weighed, or the wet gauze underneath? Was surgical dressing used, and if so, why not Surgicel powder, which would be more comparable to Arista and the oHA microparticles?

20

Response:

Thank you for pointing out this issue. We have revised the manuscript with a clear description of the rat liver injury model and porcine liver model (Lines 261-267, highlighted in red). It now reads:

25

“We initially used a rat liver volumetric defect (2 mm-diameters and 3 mm-depths) model using a biopsy punch (Fig.5a), and BICMs, Arista, Surgicel, Quikclot, and Gelfoam +T with the same weight (0.02 g) were applied onto the injury sites, and cotton gauze was used as a control (Fig.5b). To quantitatively evaluate the hemostatic performance, we measured the time to hemostasis (the time with no blood leakage, Fig.5c) and the blood loss until hemostasis (blood oozed from the materials was collected using dry, pre-weighed gauze, Fig.5d).”

30

Regarding the use of wet gauze: wet gauze is employed to moisten the exposed liver. However, during the process of creating wounds and collecting blood, dry, pre-weighed gauze is placed underneath the liver and

applied to collect the oozing blood. We apologize for the incorrect description and have revised the corresponding methodologies (Lines 481 and 542-543, highlighted in red), which now reads:

“The left liver lobe was exposed and put on wet gauze.”

“The left liver lobe was exposed, wetted with saline, and then placed on dry, weighed gauze.”

5 **Regarding the control group:** the gauze above and below the liver was dry and pre-weighed, and the overall weight gain was the amount of blood loss in the control group, demonstrating the blood loss without application of hemostats.

10 **Regarding the Surgical dressing used:** according to the manufacturer’s instructions, Fibrillar hemostat may be used precisely for continuous, multi-site bleeding in a variety of bleeding sites in open procedures, while Surgical powder can be used on broad surface oozing bleeding. Therefore, considering the volumetric injuries in the animal experiments, we selected Surgicel (Fibrillar Absorbable Hemostat) as a commercial control.

9. Line 236: I recommend reporting the hemostasis times and blood loss of the other groups here, rather than stating it is “obviously superior to other groups”.

15

Response:

Thank you for the valuable feedback, we have revised the manuscript (Lines 267-273, highlighted in red) with specific hemostasis time and blood loss of other groups. It now reads:

20 *“BICMs showed a hemostasis time of 17.5 ± 6.1 s and a blood loss of 35.0 ± 30.4 mg. Conversely, Arista, Surgicel, and Quikclot exhibited prolonged hemostasis times (45.0 ± 21.2 s, 37.5 ± 28.1 s, and 65.0 ± 84.3 s) and substantial blood loss (176.8 ± 134.6 mg, 96.2 ± 82.1 mg, and 300.2 ± 203.3 mg). Despite the rapid coagulation found in the above clotting tests, the hemostatic time (145.0 ± 50.8 s) and the blood loss (50.7 ± 186.8 mg) of Gelfoam + T exhibited considerably increased values compared to BICMs, attributable to the inadequate tissue adhesion of formed clots and the elimination of some*
25 *thrombin during blood flow.”*

10. In addition to the previous point: please clarify whether the absorbency/imbibing qualities of the hemostatic agents would significantly influence the perceived blood loss in this experiment. The blood absorbed by the hemostatic agents or dressing should still classify as total blood lost from the body in a surgical setting.

30

Response:

Thank you for pointing out the significant and insightful question. We have revised the manuscript (Lines 264-267, 547-548, and 571-572, highlighted in red) with a clear description of animal models. It now reads:

5 “To quantitatively evaluate the hemostatic performance, we measured the time to hemostasis (the time with no blood leakage, Fig.5c) and the blood loss until hemostasis (blood oozed from the materials was collected using dry, pre-weighed gauze, Fig.5d).”

“The blood oozed from the materials was collected using dry, pre-weighed gauze, and blood loss was characterized by the weight gain of gauze during hemostasis.”

“Blood oozed from the materials was collected by dry, pre-weighed gauze, and blood loss was measured by the weight gain of pre-weighed gauze.”

10 In a surgical setting, the blood absorbed by hemostatic materials should be classified as part of the total blood loss. However, given the use cases of absorbable powders, particles, and fibrous materials, quantifying their weight gain during hemostasis is difficult due to challenges in completely removing the material. Referring to literatures regarding hemostatic materials (Jiang, T. *et al.*, Li, Z. *et al.*, Tithy, L. H. *et al.*, Zhang, K. *et al.*, and Kim, K. *et al.*), we measured the weight gain of pre-weighed gauze during hemostasis as the blood loss value. Moreover, 15 we aimed to collect the blood bleeding out of the material with gauze before the blood was clotted in the material to maximize the explanation of blood loss and minimize the impact of imbibition of hemostatic materials. The control (gauze) group demonstrated total blood loss in the absence of hemostatic materials.

In summary, we thank the reviewer for identifying a critical issue regarding the evaluation of hemostatic materials. However, the comparative significance of the blood loss measured by the same method across different 20 hemostatic materials is of paramount importance, supported by our data on hemostasis time, demonstrating the superiority of BICMs in hemostasis.

References:

25 Jiang, T. *et al.* Superporous sponge prepared by secondary network compaction with enhanced permeability and mechanical properties for non-compressible hemostasis in pigs. *Nature Communications* **15**, 5460 (2024).

Li, Z. *et al.* Superhydrophobic hemostatic nanofiber composites for fast clotting and minimal adhesion. *Nature Communications* **10**, 5562 (2019).

Tithy, L. H., Rahman, A., Wong, S. Y., Li, X. & Arafat, M. T. Chitosan/starch based unoxidized tannic acid modified microparticles for rapid hemostasis with broad spectrum antibacterial activity. *Carbohydr. Polym.* **336**, 122111 (2024).

30 Zhang, K. *et al.* Gelable and Adhesive Powder for Lethal Non-Compressible Hemorrhage Control. *Advanced Functional Materials* **33**, 2305222 (2023).

Kim, K. *et al.* Coagulopathy-independent, bioinspired hemostatic materials: A full research story from preclinical models to a human clinical trial. *Science Advances* **7**, eabc9992 (2021).

11. Line 244: Coagulopathic porcine liver injury model is not clearly described. Were the oHA microparticles applied alongside dressing? Was the Surgicel dressing or powder? Were commercial hemostats used as intended (manual compression, etc)?

5

Response:

Thank you for pointing out this important issue, we have revised the manuscript accordingly (Lines 281-283 and 289-292, highlighted in red) with a clear description of the animal model and application of hemostats. It now reads:

10 *“Briefly, the coagulopathic status was induced by hemodilution, i.e., exchanging 40% of the estimated total blood volume with three times the volume of cold 0.9% saline.”*

15 *“BICMs were administered without dressing or manual compression, showing a significantly decreased time to hemostasis (within 30 s) and reduced blood loss (1.00 ± 0.41 g) until hemostasis, compared to Surgicel (232.5 ± 78.9 s and 7.60 ± 1.42 g, Fig.5g and h). For Surgicel, despite its application over a short period (~3 s) with gentle compression to position the dressing,”*

12. Please comment on whether BICMs have the risk of embolization after application?

Response:

20 Thank you for this insightful question. We have discussed the risk of embolization after BICMs' application and these results are now included in Supplementary Fig.9 and the manuscript (Lines 229-231, highlighted in red), which reads:

25 *“The improved mechanical strength and tissue adhesion of the fortified clots prevented their detachment from the bleeding site, indicating a reduced risk of microparticle embolization (Supplementary Fig.9).”*

Given the limited adhesion and cohesion, the naturally formed clots/thrombus can detach from the vessel wall and travel through the circulatory system, causing severe health problems (Sung, Y. K. *et al.*). Moreover, the powder/particle hemostats are typically to be used carefully because the potential for embolization and death may exist. For example, Kheirabadi, B. S. and colleagues reported that the residues of WoundStat were embolized in
30 the venous circulation and were trapped in the lung with associated thrombosis.

However, BICMs form a fortified clot with improved mechanical strength and tissue adhesion, preventing detachment from the bleeding site with a low potential risk of microparticle embolization. To validate the low embolic risk of BICMs, we applied ICG-labeled BICMs to a rat liver laceration model and observed the distribution of fluorescence in the rat body. As shown in Response Figure 6a, the bleeding site exhibited strong fluorescence, while no obvious fluorescence signals were identified in other parts of the rat. The organs distal to the vessels were inspected according to the literature (Kheirabadi, B. S. *et al.*) for evidence of embolization by fluorescence (Response Figure 6b) and histological analysis (Response Figure 6c). Except for the liver, no significant fluorescence was observed in the major organs of the rats. Histologically, no thrombus produced by the microparticles was found in the blood vessels of the major organs.

10

Response Figure 6 | Evaluation of the embolic risks of oHA microparticles. **a**, *In vivo* imaging system (IVIS) fluorescence analysis of the same rats examined on days 0, 3, 6, and 14 after applying oHA/xHA microparticles to a liver wound. The rat to the left is a negative control (*i.e.*, no wound induction). Representative **(b)** fluorescence images and **(c)** histological images with H&E staining of *ex vivo* organs and from a control rat and a rat with oHA microparticles implanted 14 days post-surgery.

However, further comprehensive evaluations are required in the context of long-term survival experiments and angiography-based large animal experiments, allowing us to validate the risk of hemostatic materials in thrombus formation before clinical translation.

References:

- 5 Sung, Y. K., Lee, D. R. & Chung, D. J. Advances in the development of hemostatic biomaterials for medical application. *Biomaterials Research* **25**, 37 (2021).
- Kheirabadi, B. S. *et al.* Safety Evaluation of New Hemostatic Agents, Smectite Granules, and Kaolin-Coated Gauze in a Vascular Injury Wound Model in Swine. *Journal of Trauma and Acute Care Surgery* **68** (2010).

10 13. Methods section: please clarify the following:

a. Lines 328 - 330: Minor grammatical errors

Response:

15 Thank you for pointing out this issue and prompting us to revised the manuscript (Lines 370-375, highlighted in red), which now reads:

“Evaluation of the imbibition and crosslinking for a single oHA/xHA microparticle

The oHA and xHA microparticles were firstly suspended in absolute alcohol and placed on poly-D-lysine-coated glass slides. Following solvent evaporation, the microparticles were adhered to the slide. The procedure involved the initial identification of a single microparticle in a bright field, followed by the

20 *dispensation of 20 μ L of Rho-BSA solution (50 mg mL⁻¹, pH = 7.4) onto the microparticle.”*

b. Line 356: Citrated whole blood: where was the blood sourced and how was it collected?

Response:

25 Thank you for pointing out this issue. A total of 90 mL of whole blood was collected from sheep and stored in a tube containing 10 mL of sodium citrate (109 mmol L⁻¹). We have revised the manuscript (Lines 388-390, highlighted in red), which now reads:

“Citrated blood was obtained from sheep and stored in the tube with sodium citrate (109 mmol L⁻¹, blood/sodium citrate = 9/1, v/v).”

30

c. Animals used: age and sex of animals?

Response:

We would like to express our gratitude to the reviewer for bringing our attention to this important issue. We have incorporated a more detailed description of the animals used in the methods section (Lines 478-479, 540, 555, and 580, highlighted in red), which now reads:

5 “Firstly, the Sprague-Dawley (SD) rats (12 weeks, 225–275 g weight)”

 “Firstly, the male SD rats (12 weeks, 225–275 g weight)”

 “Firstly, the domestic pigs (female, 3 months, 40–50 kg weight)”

 “Firstly, male New Zealand White rabbits (6 months. 3–3.2 kg weight)”

10 Given that the tissue adhesion, biocompatibility, degradation, and hemostatic performance of materials exhibit no significant sex disparities, sex-specific analyses were not pursued in this study. However, to mitigate the potential impact of sex on the assessment of BICMs, animals of the same sex were used in each experiment. We have added a statement to each method regarding animal experiments.

15 d. Line 423: Why was a trans-well system used to evaluate in vitro biocompatibility rather than direct co-incubation of microparticles with cells?

Response:

20 Thank you for pointing out this issue. Given the crosslinking with proteins, oHA microparticles also crosslink with the medium containing fetal bovine serum. The formed composite interfered with fluorescent staining and observation. However, the trans-well system is widely used to observe the interaction between cells and materials, including the biocompatibility of materials (Zhang, W. *et al.*, Stewart, C. L. *et al.*, and Park, J. *et al.*).

References:

25 Zhang, W. *et al.* Promoting Oral Mucosal Wound Healing with a Hydrogel Adhesive Based on a Phototriggered S-Nitrosylation Coupling Reaction. *Advanced Materials* **33**, 2105667 (2021).

 Stewart, C. L., Hook, A. L., Zelzer, M., Marlow, M. & Piccinini, A. M. PLGA-PEG-PLGA hydrogels induce cytotoxicity in conventional in vitro assays. *Cell Biochemistry and Function* **42**, e4097 (2024).

 Park, J. *et al.* A Mechanically Resilient and Tissue-Conformable Hydrogel with Hemostatic and Antibacterial Capabilities for Wound Care. *Advanced Science* **10**, 2303651 (2023).

30 e. Line 443: centrifugation speed should be reported in x g rather than rpm.

Response:

Thank you for pointing out this issue, we have revised the manuscript accordingly (Lines 515-517, highlighted in red), which now reads:

5 *“The hemolysis phenomenon induced by leach liquors or control groups after incubation was determined by measuring the concentration of released hemoglobin collected by centrifuging (200 × g, 5 minutes).”*

Reviewer #2:

10

Comments:

The manuscript presents a covalently reactive hemostat made of blood-imbibing and crosslinking microparticles, engineered to rapidly form fortified clots with improved mechanical stability and tissue adhesion. The authors demonstrate how this hemostat engages with blood through imbibition, forming covalent bonds with blood proteins and the tissue matrix to facilitate prompt clot formation. Compared to commercially available hemostats, these microparticles, synthesized from pre-crosslinked hyaluronic acid (HA) using a 1,4-butanediol diglycidyl ether (BDDE) crosslinker and subsequently grafted with *o*-phthalaldehyde (OPA) groups, exhibit superior hemostatic efficiency, as shown by the reduced hemostasis time and blood loss in rat and coagulopathic pig models. Further in vivo studies in rabbits confirm the efficacy of these microparticles, showing effective prevention of rebleeding triggered by elevated blood pressure. This work presents a novel material that can potentially lead to the clinical advancement of hemostatic agents, offering robust and effective solutions for bleeding management. The study was generally well conducted, but there are several issues to be addressed for further improving the manuscript.

20

Response:

25 We are very appreciative to the reviewer for recognizing the novelty of our study and its significance to the field.

Major comments

1. In Extended Data Figure 2a, six different chemical compounds are presented; however, the manuscript discusses the conversion results for only three of these compounds. Given that all six compounds are shown,

it would be expected that results for all compounds are provided or that the rationale for selecting only three is clarified.

Response:

5 Thank you for pointing out this issue. In response, we have provided the conversion of all six compounds. These details are now included in the manuscript (Lines 77-79, highlighted in red), which now reads:

“Yet, aromatic and alkyl aldehydes, and catechol groups exhibited no significantly detectable reaction (1.28, 0.25, and 0.99%, respectively).”

Regarding the rationale for selecting only three compounds: compared to the minimal conversion (in 60
10 minutes) of aromatic aldehydes (1.28%), alkyl aldehydes (0.25%), and catechol groups (0.99%), active esters have a considerably higher conversion rate in a physiological environment. Subsequent to a reduction in the concentration of the amine moiety (from 4.0 to 1.1 equivalents), the conversions of aromatic aldehydes, alkyl aldehydes, and catechol groups were challenging to detect by High-performance liquid chromatography (HPLC). However, OPA groups demonstrated a higher conversion rate than that of active esters, alongside long-term
15 stability in an aqueous solution. Therefore, we selected OPA groups as the reactive moieties in our microparticle design.

2. The schematic illustration in Figure 1d emphasizes covalent bonding exclusively. Is it possible that the oHA
microparticles interact with blood proteins and tissue matrix via non-covalent crosslinking such as hydrogen
20 bonding and pi-pi stacking? Potential contribution of other types of chemical interactions to rapid hemostasis may need to be discussed for additional crosslinking mechanism.

Response:

Thank you for this insightful perspective regarding the confirmation of the non-covalent crosslinking. We have
25 revised the manuscript and added Supplementary Fig.6b and Fig.4b-d to explore the potential non-covalent interactions as additional crosslinking mechanisms.

Regarding the effect of non-covalent crosslinking on interactions with blood proteins: to evaluate the potential non-covalent interaction between microparticles and blood proteins, we blocked the OPA groups on oHA microparticles to generate *EtoHA* microparticles, containing benzene rings and carboxyl groups as pi-pi stacking
30 and hydrogen bonds donors, respectively, similar to oHA microparticles. We combined *EtoHA* microparticles with

blood proteins (BSA, 50 mg mL⁻¹) to form an *EtoHA@BSA* composite (Response Figure 7). The composite was difficult to clamp and quickly dispersed in water. In contrast, the *oHA@BSA* composite exhibits a robust nature, maintaining its shape in water without microparticles leakage. This robustness is attributed to the covalent interaction provided by OPA groups. These findings were included in Supplementary Fig.6b and the manuscript (Lines 107-111, highlighted in red), which now reads:

“After the blockage of OPA groups on oHA microparticles via ethanolamine, these microparticles (EtoHA) exhibit limited crosslinking with proteins and susceptibility to fragmentation (Supplementary Fig.6b). These results collectively suggests that proteins crosslinked on the oHA microparticle surface through covalent bonding rather than physical interactions (e.g., hydrogen bonding or π - π stacking).”

Response Figure 7 | An *oHA@BSA* and an *EtoHA@BSA* composite produced by mixing *oHA* and *EtoHA* microparticles with Rho-BSA solution (50 mg mL⁻¹).

Regarding the effect of non-covalent crosslinking on interacting with tissue matrix: the abundant diversity of noncovalent bonds offers enormous design space to produce instant adhesion. However, noncovalent adhesion has low adhesion energy in wet environments, about 1–10 J m⁻² (Rose, S. *et al.* and Wang, Y. *et al.*). To assess the effect of non-covalent interaction between microparticles and the tissue matrix on adhesion, we performed hydraulic pressure, burst pressure, and lap shear tests with *EtoHA* microparticles (Response Figure 8). Similar to xHA microparticles, the adhesion performance of blood clots formed with the incorporation of *EtoHA* microparticles is diminished, exhibiting a significant reduction compared to naturally formed blood clots and *oHA@Blood* clots.

These details were included in Fig.3b-d.

Response Figure 8 | Quantification of the tissue adhesion. Lap shear tests (a), burst pressure tests (b), and tolerant hydraulic pressure tests (c) of natural blood clots, and blood clots formed based on xHA, EtoHA and oHA microparticles, as well as the indicated commercially available hemostats following incubation for 5 minutes on fresh porcine skin ($n = 4$). Values were represented as means \pm s.d., and all statistical analyses were performed using an unpaired two-tailed Student t -test. * $p < 0.05$, ** $p < 0.01$, *** $p < 0.001$, **** $p < 0.0001$.

Collective results indicate that covalent crosslinking, facilitated by OPA groups, operates directly in the interaction between microparticles and blood proteins, as well as the tissue matrix. Concurrently, non-covalent interactions of microparticles, including carboxyl groups on the hyaluronic acid backbone and the pi-pi stacking provided by benzene rings, play a limited role in the formation of blood clots and adhesion.

References:

Rose, S. *et al.* Nanoparticle solutions as adhesives for gels and biological tissues. *Nature* **505**, 382-385 (2014).
 Wang, Y. *et al.* Instant, Tough, Noncovalent Adhesion. *ACS Applied Materials & Interfaces* **11**, 40749-40757 (2019).

3. The rationale behind the optimization of the oHA microparticles is vague. The author's selection of 100 μm diameter with a 13.5% pre-crosslinking density lacks justification, especially considering that other variables showed higher imbibition rates. Further explanation of the author's choice of these parameters is required.

Response:

Thank you for highlighting this significant issue. We have included a comparison of the imbibition ratio of various microparticles in Fig.1f and g, and the manuscript.

Regarding the pre-crosslinking density: as shown in Response Figure 5a, the microparticles with a 13.5% pre-crosslinking density demonstrated a significantly higher imbibition ratio (15.34 g g^{-1}) than that with a 16.5% pre-crosslinking density (12.12 g g^{-1} , $p < 0.001$). However, the microparticles with a 13.5% pre-crosslinking density

exhibit no effect on a significant difference in imbibition rate ($p > 0.1$). Therefore, we chose microparticles with a 13.5% pre-crosslinking density for further optimization. These data were included in the manuscript (Lines 119-121, and 125-127, highlighted in red), which now reads:

5 *“We measured the weight gain of microparticles upon exposure to PBS for 10 s (defined as imbibition rate) and following saturation (defined as imbibition ratio).”*

“However, further increases in pre-crosslinking density result in the limited imbibition capability of oHA microparticles following saturation with reduced imbibition ratio, consistent with the findings reported in the literature³⁸.”

10 **Regarding the size of microparticles:** as presented in Response Figure 5b, upon decreasing the size of oHA microparticles (13.5% pre-crosslinking density), the microparticles with the size of 75-120 μm had a higher imbibition rate than that with larger size ($p < 0.05$ with 120-180 μm and $p < 0.01$ with 180-325 μm), similar to the microparticles with the size of 38-75 μm ($p > 0.1$). These data were included in the manuscript (Lines 132-133, highlighted in red), which now reads:

15 *“However, further reducing the size of oHA microparticles caused a limited enhancement in imbibition speed, alongside a decline in the imbibition ratio.”*

Considering the imbibition rate and ratio, we selected the microparticles with a size of 75-120 μm and a 13.5% pre-crosslinking density as BICMs in further study. These details were included in the manuscript (Lines 133-135, highlighted in red), which now reads:

20 *“Therefore, in this study, we adopted a pre-crosslinking density of 13.5% and a diameter of 75~120 μm for oHA microparticles by considering the fast and high imbibition.”*

4. Emphasizing and presenting additional data on tissue adhesion more thoroughly would strengthen the manuscript. While the hydraulic pressure test in Figure 3b provides valuable insight, additional experiments, such as a lap shear test, could further support the suggested advantages of the oHA microparticles in tissue
25 adhesion.

Response:

Thank you for the insightful suggestion regarding the additional data on tissue adhesion. These details were included in Fig.3b-d and the manuscript (Lines 209-212, highlighted in red), which now reads:

30 *“To quantitatively evaluate the adhesion performance of formed clots, we employed the standard lap shear test and burst pressure test (Fig.3b and c). Compared to natural blood clots and clots formed using*

other materials, the oHA@Blood clot adhered on porcine skin demonstrated the highest shear strength (8.53 ± 0.67 kPa) and burst pressure (266.46 ± 51.04 mmHg).”

5. In Figure 4a, it is recommended to quantify the LIVE/DEAD staining data to clarify the biocompatibility of the microparticles more effectively. In Figure 4e, in vivo degradation profile of the implanted oHA microparticles needs to be quantified and compared with that of xHA microparticles.

Response:

Thank you for this insightful comment. We have revised Fig.4a and expanded Supplementary Fig.14 with supplementary quantification.

Regarding the LIVE/DEAD staining data: we quantified cell viability by calculating the number of live (green fluorescence) and dead (red fluorescence) cells using ImageJ (Response Figure 9). These details were included in Fig.4a and the manuscript (Lines 241-243, highlighted in red), which now reads:

“Both LIVE/DEAD staining (Fig.4a) and CCK-8 assays (Supplementary Fig.12) revealed that oHA microparticles showed no obvious cytotoxicity ($> 90\%$ cell viability) towards L929 cells, which was comparable to xHA microparticles.”

Response Figure 9 | Cell viability of mouse fibroblast (L929) cells assessed using a LIVE/DEAD assay after 24 hours of co-incubation with or without xHA/oHA microparticles (20 mg). Values were represented as means \pm s.d. ($n = 5$), and all statistical analyses were performed using an unpaired two-tailed Student t-test. * $p < 0.05$, ** $p < 0.01$, *** $p < 0.001$, **** $p < 0.0001$.

Regarding the in vivo degradation profile: considering the deterioration in quality of microparticles during the collection process and the occurrence of tissue infiltration, the residual volume of the implant was used as the quantitative standard. As shown in Response Figure 10, the volume of the implant gradually decreases. Specifically, the degradation of oHA microparticles is slightly slower than xHA microparticles, but there is no significant difference. These results were included in Supplementary Fig.14 and the manuscript (Lines 252-255, highlighted in red), which now reads:

“Notably, the oHA and xHA microparticles exhibited gradual biodegradation, characterized by a comparable and progressive reduction in volume over the 3-months implantation period (Fig.4e and Supplementary Fig.14), which enables hemostats to avoid the need for secondary removal⁴⁷.”

5 **Response Figure 10 | Residual volume of xHA and oHA microparticles in rat subcutaneous implantation model.** All values represent mean \pm s.d. ($n = 3$). All statistical analyses were performed using an unpaired two-tailed Student *t*-test. * $p < 0.05$, ** $p < 0.01$, *** $p < 0.001$, **** $p < 0.0001$.

6. Although immunostaining for macrophages was performed in Extended Data Figure 10, the possibility of immunogenicity and inflammation should be more carefully investigated given the use of BDDE crosslinker.
10

Response:

Thank you for this suggestion regarding the potential immunogenicity and inflammation risks associated with the use of 1,4-butanediol diglycidyl ether (BDDE) crosslinker. We have revised the manuscript and Supplementary Fig.10 to illustrate the potential safety related to the use of BDDE crosslinker.
15

BDDE is biodegradable, with minimal toxicity compared to other ether-bond crosslinking agents (De Boule, K. *et al.*). However, Jeong *et al.* demonstrated *in vitro* that BDDE is toxic above 100 ppm. Additionally, while BDDE is a potential inducer of late-onset immune reactions (Wojtkiewicz, M. *et al.*) and the primary cause of occupational allergic contact dermatitis in certain cases (*e.g.*, Jolanki and colleagues have reported that BDDE was the main allergen in brush factory workers), in clinical studies, *in vivo* BDDE pre-crosslinked HA has significant anti-inflammatory activity and reduces immune responses (Andrade del Olmo, J. *et al.* and Dewangan, R. *et al.*).
20

To limit toxicity, potential immunogenicity, and inflammation of the BDDE crosslinker, we applied a step to effectively remove BDDE residuals. The method involves repeated dialysis and dehydration, enabling the unreacted crosslinking agent to diffuse out of the microparticles (depicted in the Supplementary Information).

Given that the FDA-approved residual crosslinker concentration is below 2 ppm (De Boule, K. *et al.*), we degraded the microparticles and determined the residual BDDE by fluorospectrophotometry (depicted in the Supplementary Information). Both microparticles exhibited negligible residual BDDE crosslinker (100 mg of microparticles contained below 0.8 ug of BDDE residuals, Response Figure 11). These findings indicate that the BDDE can be safely used to prepare hemostatic microparticles without irritation of immunogenicity and inflammation. These results were included in the Supplementary Fig.10 and the manuscript (Lines 238-239, highlighted in red), which now reads:

“Residual BDDE crosslinker was rinsed thoroughly, in accordance with the FDA approval (< 2 ppm, Supplementary Fig.10).”

Response Figure 11 | Quantification of residual BDDE crosslinkers. a, Fluorescence emission spectra of the mixture with varying concentrations of BDDE. b, Calibration curve of BDDE. c, Fluorescence emission spectra of xHA and oHA microparticles (100 mg) enzymatic products characterized by fluorospectrophotometry.

References:

De Boule, K. *et al.* A Review of the Metabolism of 1,4-Butanediol Diglycidyl Ether-Crosslinked Hyaluronic Acid Dermal Fillers. *Dermatologic Surgery* **39** (2013).

Jeong, C. H. *et al.* In vitro toxicity assessment of crosslinking agents used in hyaluronic acid dermal filler. *Toxicology in Vitro* **70**, 105034 (2021).

Wojtkiewicz, M. *et al.* Are We Overlooking Harms of BDDE-Cross-Linked Dermal Fillers? A Scoping Review. *Aesthetic Plastic Surgery* (2024).

Jolanki, R., Estlander, T. & Kanerva, L. Contact allergy to an epoxy reactive diluent: 1,4-butanediol diglycidyl ether. *Contact Dermatitis* **16**, 87-92 (1987).

Andrade del Olmo, J. *et al.* Drug Delivery from Hyaluronic Acid–BDDE Injectable Hydrogels for Antibacterial and Anti-Inflammatory Applications. *Gels* **8**, 223 (2022).

Dewangan, R. *et al.* In-Vitro Biocompatibility Determination of Bladder Acellular Matrix Graft. *Trends in biomaterials & artificial organs* **25**, 161-171 (2011).

7. The minor injury created in the pig model may limit the assessment of the hemostat's effectiveness, especially given the choice of a larger animal model intended to validate its function. Similarly, the injury applied in the rat model may also be too minimal to thoroughly evaluate the hemostatic properties. Larger injuries might provide a more rigorous test of the hemostat's performance and better support its potential clinical applicability.

5

Response:

Thank you for this valuable suggestion.

The porcine liver injury model is a widely employed large-animal model to evaluate the efficacy of topical hemostats (Jiang, T. *et al.*, Yuk, H. *et al.*, and Eskildsen, M. P. R. *et al.*). Moreover, we developed a hemodilution-induced coagulopathic model (Kim, K. *et al.*) to assess BICMs' capability for hemostasis independent of coagulation. In this model, plasma fibrinogen is rapidly reduced proportionally to hemodilution and meanwhile fibrin clots are more prone to fibrinolysis because major antifibrinolytic proteins are decreased (Bolliger, D., Görlinger, K. and Tanaka, K. A.).

Despite the reduced level of hemorrhage in this model compared to other injury models in the literature, the lower blood loss and shorter hemostatic time of BICMs in comparison to Surgicel demonstrate the hemostatic efficacy and its potential for use in clinical scenarios, especially in conditions of coagulation disorders.

Likewise, the rat liver volumetric injury model is employed to evaluate the hemostatic efficacy of hemostats (Jiang, T. *et al.*, Yang, Y. *et al.*, Yuk, H. *et al.*, Hu, S. *et al.*, and Bao, G. *et al.*). This extensively employed and standardized hemorrhage model will enable a comprehensive evaluation of the hemostatic efficacy of BICMs compared to other commercially available hemostatic materials.

References:

Jiang, T. *et al.* Superporous sponge prepared by secondary network compaction with enhanced permeability and mechanical properties for non-compressible hemostasis in pigs. *Nature Communications* **15**, 5460 (2024).

Yuk, H. *et al.* Rapid and coagulation-independent haemostatic sealing by a paste inspired by barnacle glue. *Nature Biomedical Engineering* (2021).

Eskildsen, M. P. R. *et al.* An autologous blood-derived patch as a hemostatic agent: evidence from thromboelastography experiments and a porcine liver punch biopsy model. *Journal of Materials Science: Materials in Medicine* **34**, 20 (2023).

Kim, K. *et al.* Coagulopathy-independent, bioinspired hemostatic materials: A full research story from preclinical models to a human clinical trial. *Science Advances* **7**, eabc9992 (2021).

Bolliger, D., Görlinger, K. & Tanaka, K. A. Pathophysiology and treatment of coagulopathy in massive hemorrhage and hemodilution. *Anesthesiology* **113**, 1205-1219 (2010).

Yang, Y. *et al.* An Injectable Hydrogel with Ultrahigh Burst Pressure and Innate Antibacterial Activity for Emergency Hemostasis and Wound Repair. *Advanced Materials* **36**, 2404811 (2024).

Hu, S. *et al.* An All-in-One "4A Hydrogel": through First-Aid Hemostatic, Antibacterial, Antioxidant, and Angiogenic to Promoting Infected Wound Healing. *Small* **19**, 2207437 (2023).

5 Bao, G. *et al.* Liquid-infused microstructured bioadhesives halt non-compressible hemorrhage. *Nature Communications* **13**, 5035 (2022).

8. The oHA microparticles outperformed control hemostats (Arista, Surgicel) in hemostasis tests in Figure 5 and 6, but I'm not sure whether they are the best controls to be compared with the oHA microparticles. Are there any rationales to choose such control hemostats for comparison?

10

Response:

Thank you for this valuable suggestion.

15 Arista (AH Absorbable Hemostat) is an adjunctive hemostatic device intended to assist in situations where the control of capillary, venous, and arteriolar bleeding by pressure, ligature, and other conventional procedures are ineffective or impractical. Studies in animal models and preclinical trials indicate the broad potential application of Arista in surgical fields, including laparoscopic or open surgeries (Di Benedetto, M. *et al.*; LyBarger, K. S.), and in the literature as a commercially available control (Yang, Y. *et al.*; Liu, C. *et al.*; Ito, S. *et al.*).

20 Surgicel (FIBRILLAR Absorbable Hemostat), a widely used local hemostat, is designed to halt continuous oozing in various bleeding sites, and can be separated and manipulated to ensure precise placement around bleeding sites (Schonauer, C., Tessitore, E., Barbagallo, G., Albanese, V. and Moraci, A.). Moreover, oxidized cellulose (*i.e.*, Surgicel® hemostatic products) is always selected as a commercially available control (He, H. *et al.*; Zheng, Y. *et al.*; Haghniaz, R. *et al.*).

Given the inherent properties of being absorbable and blood imbibition, alongside wide applications, we chose these two topical hemostats as commercially available controls.

25 However, in accordance with the recommendation of Reviewer #1's, we expanded the comparison with Quikclot (Advanced Clotting Sponge) and thrombin-soaked Gelfoam (absorbable gelatin compressed sponge) in both *in vitro* and *in vivo* experiments. These additional analyses demonstrate the hemostatic efficacy of BICMs.

References:

30 Di Benedetto, M. *et al.* The use of Arista AH as a local haemostatic agent in distal splenopancreatectomy: report of two cases. *Drugs Context* **13** (2024).

LyBarger, K. S. Review of Evidence Supporting the Arista™ Absorbable Powder Hemostat. *Med Devices (Auckl)* **17**, 173-188 (2024).

Yang, Y. *et al.* A Laparoscopically Compatible Rapid-Adhesion Bioadhesive for Asymmetric Adhesion, Non-Pressing Hemostasis, and Seamless Seal. *Advanced Healthcare Materials* **13**, 2304059 (2024).

Liu, C. *et al.* A highly efficient, in situ wet-adhesive dextran derivative sponge for rapid hemostasis. *Biomaterials* **205**, 23-37 (2019).

5 Ito, S. *et al.* Improved hydration property of tissue adhesive/hemostatic microparticle based on hydrophobically-modified Alaska pollock gelatin. *Biomaterials Advances* **159**, 213834 (2024).

Schonauer, C., Tessitore, E., Barbagallo, G., Albanese, V. & Moraci, A. The use of local agents: bone wax, gelatin, collagen, oxidized cellulose. *European Spine Journal* **13**, S89-S96 (2004).

10 He, H. *et al.* Efficient, biosafe and tissue adhesive hemostatic cotton gauze with controlled balance of hydrophilicity and hydrophobicity. *Nature Communications* **13**, 552 (2022).

Zheng, Y. *et al.* Hemostatic patch with ultra-strengthened mechanical properties for efficient adhesion to wet surfaces. *Biomaterials* **301**, 122240 (2023).

Haghniaz, R. *et al.* Injectable, Antibacterial, and Hemostatic Tissue Sealant Hydrogels. *Advanced Healthcare Materials* **n/a**, 2301551 (2023).

15 Minor comments

1. On page 5, line 118, "Supplementary Information" needs to specify figure number.

Response:

20 Thank you for the reminder, we have supplied the specific figure number in the manuscript (Lines 128-129, highlighted in red), stating:

“Upon decreasing the size of oHA microparticles (13.5% pre-crosslinking density) through changing conditions in preparation (Supplementary Fig. 1b and c),”

2. On page 6, the reference to Figure 2c is missing in the text.

25

Response:

Thank you for the reminder. We apologize for our negligence in referring to Fig.2c and have revised the manuscript (Lines 146-148, highlighted in red), stating:

30 *“To further investigate the effect of OPA groups on clot formation, we varied oHA microparticles with different densities of OPA groups (Fig.2c).”*

3. On page 6, line 143, the reference to Figure 3c is incorrect, which should be corrected to Figure 2d.

Response:

Thank you for the reminder. We apologize for our negligence in referring to Fig.2d and have revised the manuscript (Lines 153-154, highlighted in red), stating:

“Furthermore, the clot withstood a weight of 250 g without being squashed (~ 31.2 kPa, Fig.2d),”

5

4. In the “Rapid hemostasis of liver injury in rat and coagulopathic pig models” section, the figures are incorrectly labeled as Figure 4b-d and 4f-h, which should be corrected to Figure 5b-d and 5f-h. Also, the reference to Figure 5a is missing.

10 **Response:**

Thank you for the reminder. We apologize for our negligence in referring to Fig.4b-d and f-h, as well as the omission of reference to Fig.5a. We have revised the manuscript (In the “Rapid hemostasis of liver injury in rat and coagulopathic pig models” section, highlighted in red) to ensure that all figures have been cited correctly.

- 15 5. While the time point in Extended Data Figure 2 is mentioned as 200 minutes, the manuscript text refers to “three hours,” creating a discrepancy that should be corrected for consistency.

Response:

20 Thank you for your careful consideration. In the original manuscript, we dissolved and stored compounds in phosphate Buffered Saline (PBS) for defined periods of time (30 days for OPA, and 3 hours for NHS and PFP esters, respectively). Subsequently, we incubated those reactive compounds with the amine moiety for 200 minutes, and measured the conversion of the reactive compounds. However, we recognized this could be misinterpreted as a discrepancy. To overcome any potential misinterpretation, we revised Supplementary Fig.2e and the Supplementary Information with a clear description of our methodology.

Reviewer #1

Comment:

My previous concerns have been addressed.

5 **Response:**

We are sincerely grateful for the reviewer's positive feedback on our previous responses.

Comment:

One minor question was why sheep blood was used for the rheology testing.

10 **Response:**

We used sheep blood for rheology testing due to its availability in large quantities, and its close resemblance to human blood in terms of coagulation and rheological properties, which are essential for obtaining biologically relevant results. Siller-Matula, J. M. et al. reported that sheep exhibit clotting times most similar to those of humans, making them a suitable model for translational coagulation studies. Additionally, sheep blood demonstrates a high degree of similarity to human blood in terms of viscosity (Ecker, P. et al.; Windberger, U. et al.). This selection ensures the reliability and translational relevance of our findings to human physiology.

References:

Siller-Matula, J. M., Plasenzotti, R., Spiel, A., Quehenberger, P. & Jilma, B. Interspecies differences in coagulation profile. *Thromb. Haemost.* **100**, 397-404 (2008).

20 Ecker, P. et al. Animal blood in translational research: How to adjust animal blood viscosity to the human standard. *Physiol Rep* **9**, e14880 (2021).

Windberger, U., Bartholovitsch, A., Plasenzotti, R., Korak, K. J. & Heinze, G. Whole Blood Viscosity, Plasma Viscosity and Erythrocyte Aggregation in Nine Mammalian Species: Reference Values and Comparison of Data. *Exp. Physiol.* **88**, 431-440 (2003).

25 **Comment:**

Additionally, the text should be thoroughly cleaned up throughout before publishing. There are many minor english mistakes and areas for improvement, and clean-up by by an experienced science editor is needed.

Response:

30 Thank you for your insightful comments. We have carefully addressed the language-related concerns with the assistance of an experienced editor to improve the overall quality and clarity of the manuscript.

Comment:

For example, in the abstract, providing quantitative results rather than saying 'outstanding' would be better.

Response:

Thank you for your insightful consideration. We have incorporated the quantitative results and removed the exaggerated language in the abstract, which now reads:

“In contrast to commercial hemostats, the microparticles achieve rapid hemostasis (within 30 seconds) and significantly reduce blood loss (approximately 35 mg and 1 g in the rat and coagulopathic pig models, respectively),”

Comment:

Another example, is that fibrin is referred to as 'fibrins'. There are many other examples.

Response:

Thank you for highlighting the issue. We sincerely apologize for the incorrect use of 'fibrin' and have revised the manuscript accordingly (line 51), which now reads:

“In the final phase of coagulation, active coagulation factor XIII (FXIIIa) mediates covalent crosslinking between fibrin fibers, as well as between fibrin fibers and the tissue matrix.”

Reviewer #2:

Comments:

The authors have addressed all issues raised by this reviewer. Accordingly, the issues are now mostly resolved, and the manuscript has been improved. I do not have any other comments to the authors.

Response:

We sincerely appreciate the reviewer's kind remarks regarding our previous responses.